# JODI: UNIFICATION OF VISUAL GENERATION AND UNDERSTANDING VIA JOINT MODELING

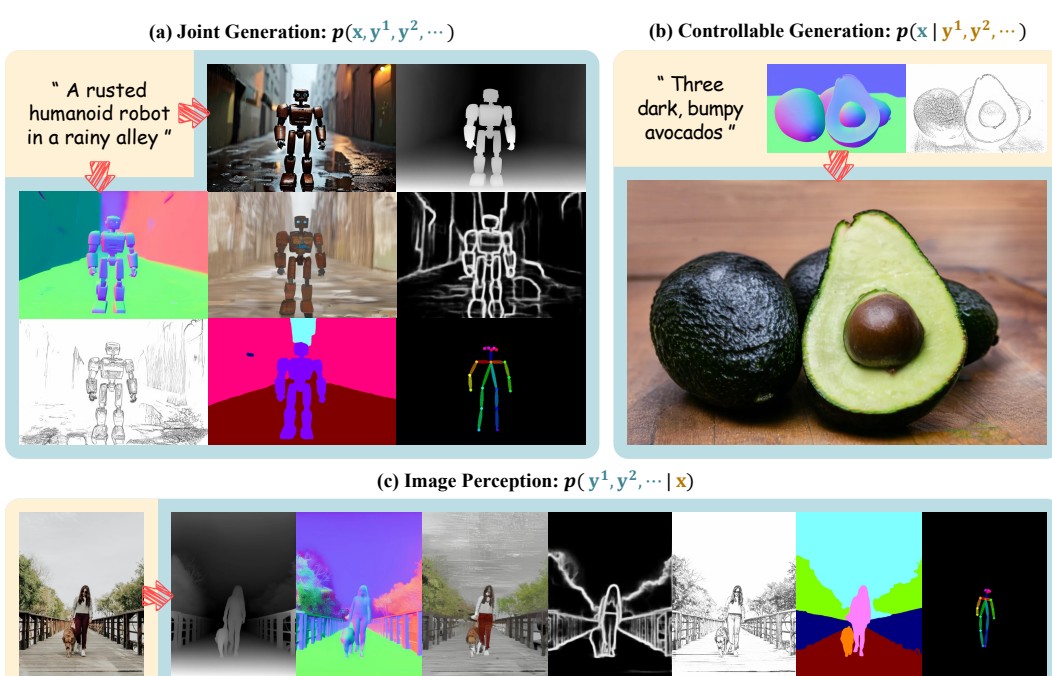

Figure 1: Our Jodi framework is capable of performing (a) joint generation, (b) controllable generation, and (c) image perception in a unified diffusion model. More visual results can be found in the appendix.

## ABSTRACT

Visual generation and understanding are two deeply interconnected aspects of human intelligence, yet they have been traditionally treated as separate tasks in machine learning. In this paper, we propose Jodi, a diffusion framework that unifies visual generation and understanding by jointly modeling the image domain and multiple label domains. Specifically, Jodi is built upon a linear diffusion transformer along with a Role-Switch mechanism, which enables it to perform three particular types of tasks: (1) joint generation, where the model simultaneously generates images and multiple labels; (2) controllable generation, where images are generated conditioned on any combination of labels; and (3) image perception, where multiple labels can be predicted at once from a given image. Furthermore, we present the Joint-1.6M dataset, which contains 200K high-quality images collected from public sources, automatic labels for 7 visual domains, and LLM-generated captions. Extensive experiments demonstrate that our Jodi excels in generation tasks and performs competitively in understanding tasks. Besides, Jodi exhibits strong extensibility to new visual domains. Codes, data, and model weights will be publicly available.

## 1 INTRODUCTION

Visual generation and understanding have long been regarded as two separate research fields, each addressed by specialized models. However, from the perspective of human cognition (Ellamil et al., 2012; Kozbelt, 2001; Chamberlain et al., 2019; Fernandes et al., 2018), a profound understanding of a visual scene/object is fundamental to its creation; conversely, the process of creating that scene/object can further enhance and refine our understanding of it. In other words, generation and understanding are two sides of the same coin and deeply interdependent. Therefore, exploring the *unification of visual generation and understanding* within a single foundation model, analogous to the human brain, might be a promising avenue toward human-level artificial intelligence.

Theoretically, generation and understanding can be associated through the joint distribution. Let $\mathbf{x}$ denote the image domain and $\mathbf{y}$ denote the label domain, generation tasks are typically formulated as learning $p(\mathbf{x})$ for unconditional generation and $p(\mathbf{x} \mid \mathbf{y})$ for conditional generation, whereas understanding tasks are commonly represented as $p(\mathbf{y} \mid \mathbf{x})$. It is a theoretical fact that, once we have the joint distribution $p(\mathbf{x}, \mathbf{y})$, we can derive any of the corresponding marginal distributions $p(\mathbf{x})$ and $p(\mathbf{y})$, as well as the conditional distributions $p(\mathbf{x} \mid \mathbf{y})$ and $p(\mathbf{y} \mid \mathbf{x})$[1]. This implies that the joint distribution inherently encodes the interdependence between generation tasks and understanding tasks. Inspired, an intriguing idea arises: Is it possible to achieve the unification of visual generation and understanding by *jointly modeling the image domain and the label domain*?

In this paper, we propose **Jodi** (**Jo**int **Di**ffusion), a diffusion model that jointly learns the distributions over the image domain $\mathbf{x}$ and multiple label domains $\mathbf{y}^1, \mathbf{y}^2, \ldots$, including depth, normal, albedo, edge, line art, segmentation, and human skeleton. During the training process, a Role-Switch mechanism will assign each domain to one of three roles: as a generation target, as a condition input, or to be ignored. As a result, our unified model simultaneously learns three types of probability distributions, including: 1) $p(\mathbf{x}, \mathbf{y}^1, \mathbf{y}^2, \cdots)$, *joint generation*, where the model simultaneously generates both the image and the corresponding labels of different domains; 2) $p(\mathbf{x} \mid \mathbf{y}^1, \mathbf{y}^2, \cdots)$, *controllable generation*, where the images are generated conditioned on any combination of the label domains; 3) $p(\mathbf{y}^1, \mathbf{y}^2, \cdots \mid \mathbf{x})$, *image perception*, where the model accepts an input image and predicts multiple labels at once. In a word, the proposed model is capable of performing both image generation and understanding, as shown in Figure 1.

To effectively capture the correspondence and model the consistency among different visual domains, we employ the powerful attention mechanism (Vaswani et al., 2017; Peebles & Xie, 2023). However, as the number of domains increases, the computational burden of full attention grows quadratically in terms of both time and space, making the training inefficient or even infeasible. To address this issue, we adopt the linear diffusion transformer (Katharopoulos et al., 2020; Xie et al., 2025a) and design a masked variant to accommodate our Role-Switch mechanism, which achieves linear time and space complexities relative to the number of domains. To further enhance the inter-domain consistency, we introduce domain-invariant positional embeddings to provide an explicit cue for the spatial alignment between visual domains. As a result, our framework is capable of modeling as many as 8 visual domains simultaneously with high consistency.

Our contributions are summarized below:

1. Inspired by the theoretical fact that the joint distribution intrinsically connects generation and understanding, we propose to jointly model the image domain and multiple label domains, achieving a *unification of visual generation and understanding*. As a result, our framework is capable of joint generation, controllable generation, and image perception in a unified diffusion model.

2. Our model effectively captures complex inter-domain relationships through the masked linear attention, and achieves high consistency across different domains by using the proposed domain-invariant positional embeddings.

3. Our model supports novel applications, including joint generation of images and labels, and performing multiple understanding tasks at the same time. Besides, our model can simultaneously handle as many as 8 visual domains, and can be easily extended to more new domains.

---

[1] $p(\mathbf{x}) = \int p(\mathbf{x}, \mathbf{y}) \, \mathrm{d}\mathbf{y}$, $p(\mathbf{x} \mid \mathbf{y}) = p(\mathbf{x}, \mathbf{y}) \, / \, p(\mathbf{y})$, $p(\mathbf{y} \mid \mathbf{x}) = p(\mathbf{x}, \mathbf{y}) \, / \, p(\mathbf{x})$

## 2 RELATED WORK

**Diffusion Models for Image Generation**   Diffusion models have made remarkable progress in image generation (Sohl-Dickstein et al., 2015; Song & Ermon, 2019; Ho et al., 2020; Song et al., 2021; Lipman et al., 2023; Liu et al., 2023; Albergo & Vanden-Eijnden, 2023), with large-scale text-to-image (T2I) models excelling in generating both photorealistic and imaginative scenes (Rombach et al., 2022; Saharia et al., 2022; Balaji et al., 2022; Li et al., 2024; Chen et al., 2024a; Esser et al., 2024; BlackForestLab, 2024; Xie et al., 2025a). To enhance the controllability, conditional diffusion models introduce spatial conditions to enable more fine-grained control over the generated images (Zhang et al., 2023; Mou et al., 2024; Tan et al., 2024; Zhang et al., 2025b). Moreover, several studies (Li et al., 2023b; Wang et al., 2024c; Zhou et al., 2024; 2025b;c) introduce layout information (e.g., bounding boxes or masks) for each instance to improve multi-instance generation. Subsequent methods further improve the efficiency by unifying different types of conditions within a single model (Zhao et al., 2024; Qin et al., 2024; Xu et al., 2025b).

**Diffusion Models for Image Understanding**   Diffusion models have also exhibited superior performance in image understanding tasks, such as geometry estimation (Ke et al., 2024; Fu et al., 2024; Lee et al., 2024; Ye et al., 2024a; Zeng et al., 2024; Xu et al., 2025a; He et al., 2025), segmentation (Xu et al., 2023; Zhao et al., 2023; Pnvr et al., 2023; Zhu et al., 2024), and edge detection (Ye et al., 2024b). These methods either use pretrained diffusion models as feature extractors or reformulate the prediction objectives with diffusion frameworks. Furthermore, several works (Wang et al., 2023; Zhao et al., 2025) unify a wide range of understanding tasks into a single diffusion model, demonstrating the capability of diffusion models in complicated image understanding.

**Diffusion Models for General Purposes**   Recent efforts (Lin et al., 2025; Xiao et al., 2024; Le et al., 2025; Chen et al., 2024b; Fu et al., 2025; Li et al., 2025a) have developed generalist diffusion models to handle various tasks of both image generation and understanding within a single model. Typically, these methods achieve general capabilities by training diffusion models on large-scale datasets that span diverse visual tasks. However, they do not investigate the relationships among different tasks, and each task requires a separate inference process. In contrast, our work emphasizes and models the correspondence and consistency among various visual domains (tasks), enabling novel applications unattainable with previous generalist methods, such as joint generation of image and multiple labels for data synthesis, and simultaneously performing multiple understanding tasks.

Several works (Zhang et al., 2024; Byung-Ki et al., 2025; Wang et al., 2025) have also incorporated multiple visual domains into a single model. However, these approaches are constrained by either the number of domains or image resolution. In contrast, our method incorporates as many as 8 visual domains with image resolutions of approximately 1024×1024 pixels, making it significantly more versatile in real-world applications.

**Multi-modal Generation and Understanding**   In the context of multi-modal learning, previous works have explored the unification of vision and language by modeling images and texts with autoregressive (Lu et al., 2023; Team, 2024; Wang et al., 2024b; Wu et al., 2024; 2025; Chen et al., 2025b), diffusion (Bao et al., 2023; Li et al., 2025b; Yang et al., 2025), or hybrid frameworks (Zhou et al., 2025a; Xie et al., 2025b; Ma et al., 2024; Deng et al., 2025; Xie et al., 2025c). These methods are capable of various cross-modality tasks, such as image-text mixed generation, text-to-image generation, and visual question-answering. In contrast to these methods that focus on unifying vision and language modalities, our work concentrates on the unification of pure visual domains within a single diffusion framework.

## 3 METHOD

**Overview**   In this section, we present the details of our Jodi framework, which unifies visual generation and understanding within a single diffusion model by jointly modeling the image domain and multiple label domains. As shown in Figure 2, our Jodi mainly consists of three parts: a Deep Compression Autoencoder (DC-AE) (Chen et al., 2025a), a Role-Switch mechanism, and a linear diffusion transformer backbone. Specifically, all of the image domain and the label domains are first compressed into a set of tokens by DC-AE with a downsampling factor of 32. Then, each domain is randomly assigned one of three roles: as a generation target, as a condition input, or to be ignored. Depending on the roles, a Switch module further processes the tokens in one of the following ways:

Figure 2: Overview of our Jodi framework. For clarity, only four domains are illustrated.

adding noise, preserving their values, or setting them to zero. Subsequently, tokens from all domains are concatenated and fed into the linear diffusion transformer, which facilitates interactions across these domains and predicts the velocity field as in Rectified Flow (Liu et al., 2023). Please refer to the appendix for more details on the framework architecture.

## 3.1 JOINT MODELING WITH ROLE-SWITCH MECHANISM

**Role Assignment**  Let $\mathbf{y}^0 = \mathbf{x}$ denote the tokens of image domain and $\mathbf{y}^1, \mathbf{y}^2, \ldots, \mathbf{y}^M$ denote the tokens of $M$ distinct label domains. At each training iteration, each domain is randomly assigned one of three roles: 1) [G], which means the model will learn to generate this domain; 2) [C], which means the model will use this domain as a condition; 3) [X], which means this domain will be ignored. In this manner, our model learns a class of probability distributions as follows:

$$p\big(\{\mathbf{y}^m \,|\, \text{role}^m = [\text{G}]\} \,\big|\, \{\mathbf{y}^m \,|\, \text{role}^m = [\text{C}]\}\big). \tag{1}$$

Since each domain can be an outcome, a condition, or be ignored in Eq. (1), our model learns diverse distributions, including three most typical ones: 1) $p(\mathbf{x}, \mathbf{y}^1, \mathbf{y}^2, \cdots)$, joint generation, where the model simultaneously generates both the image and the corresponding labels of different domains; 2) $p(\mathbf{x} \,|\, \mathbf{y}^1, \mathbf{y}^2, \cdots)$, controllable generation, where the images are generated conditioned on any combination of the label domains; 3) $p(\mathbf{y}^1, \mathbf{y}^2, \cdots \,|\, \mathbf{x})$, image perception, where the model accepts an input image and predicts multiple labels at once. In a word, our method unifies various distributions related to both image generation and understanding within a single model.

**Switch Module**  Depending on the roles assigned, the Switch module processes the tokens in different ways, as shown on the right of Figure 2. Specifically, at diffusion time step $t$, the [G] tokens are linearly interpolated with noise $\boldsymbol{\epsilon}^m \sim \mathcal{N}(\mathbf{0}, \mathbf{I})$ as in Rectified Flow (Liu et al., 2023), the [C] tokens remain unchanged, and the [X] tokens are set to zero. Let $\mathbf{y}_0^m = \mathbf{y}^m$, this process is formulated as follows:

$$\mathbf{y}_t^m = \begin{cases} (1-t)\mathbf{y}_0^m + t\boldsymbol{\epsilon}^m & \text{if role}^m = [\text{G}] \\ \mathbf{y}^m & \text{if role}^m = [\text{C}] \\ \mathbf{0} & \text{if role}^m = [\text{X}] \end{cases} \tag{2}$$

**Objective Function**  Given the processed tokens in Eq. (2), we optimize our model by flow matching (Lipman et al., 2023; Liu et al., 2023). Specifically, our model learns to predict the velocity field of [G] tokens conditioned on [C] tokens, with the following objective function:

$$\mathcal{L} = \mathbb{E}_{t \sim \pi_{\ln}(0,1), \, \boldsymbol{\epsilon}^{0:M} \sim \mathcal{N}(\mathbf{0}, \mathbf{I}), \, \mathbf{y}_0^{0:M} \sim \mathcal{D}} \left[ \sum_{m: \, \text{role}^m = [\text{G}]} \left\| \mathbf{v}_\theta^m(\mathbf{y}_t^0, \cdots, \mathbf{y}_t^M, t) - (\boldsymbol{\epsilon}^m - \mathbf{y}_0^m) \right\|^2 \right], \tag{3}$$

where $\mathbf{v}_\theta(\cdot)$ is the velocity predictor with a linear transformer architecture, introduced in Section 3.2.

## 3.2 MODEL ARCHITECTURE

**Linear Diffusion Transformer**  We employ the attention mechanism (Vaswani et al., 2017) to model the interaction among different visual domains and predict the velocity field in Equation (3). However, as the number of domains increases, we need to carefully consider the computational complexity. Suppose we have $M$ visual domains in total, each domain contains $N$ tokens, and each token is $D$-dimensional. In this setting, the full attention mechanism exhibits a time complexity of $\mathcal{O}(M^2 N^2 D + MND^2)$ and a space complexity of $\mathcal{O}(M^2 N^2 + MND)$, both scaling quadratically with respect to the number of domains $M$. In consequence, training our model with a full attention diffusion transformer (Esser et al., 2024; BlackForestLab, 2024) is computationally inefficient or even infeasible. To solve this problem, we choose Sana (Xie et al., 2025a) as our backbone, which adopts linear transformer (Katharopoulos et al., 2020) for efficient text-to-image generation. Using linear transformer, the time complexity is reduced to $\mathcal{O}(MND^2)$ and the space complexity to $\mathcal{O}(MND)$, both of which are linear with respect to $M$. As a result, our model can efficiently handle as many as 8 visual domains. An empirical comparison on the computational cost can be found in the appendix.

When a domain is assigned the role `[X]`, i.e., to be ignored, the corresponding tokens should not participate in the attention computation. To this end, we design a masked version of linear attention. For a single attention head, let $Q_i, K_i, V_i \in \mathbb{R}^{1 \times d}$ denote the query, key, and value of the $i^{\text{th}}$ token, and $m_i \in \{0, 1\}$ indicate whether to ignore this token, the masked linear attention is designed as:

$$O_i = \frac{\text{ReLU}(Q_i)\left(\sum_{j=1}^{MN} \text{ReLU}(m_j K_j)^T V_j\right)}{\text{ReLU}(Q_i)\left(\sum_{j=1}^{MN} \text{ReLU}(m_j K_j)^T\right)}, \quad i = 1, 2, \ldots, MN. \tag{4}$$

When $m_j = 0$ in Eq. (4), the $j^{\text{th}}$ token vanishes from both the denominator and numerator, which means it is excluded from the attention computation.

**Domain-invariant Positional Embeddings**  A notable feature of our backbone Sana (Xie et al., 2025a) is that it does not use explicit positional embeddings (NoPE) (Haviv et al., 2022; Kazemnejad et al., 2023). However, in our multi-domain scenario, there is a strong spatial correspondence between the visual domains. Therefore, it is necessary to explicitly indicate the spatial positions to facilitate precise spatial alignment across different domains. To this end, we add domain-invariant sinusoidal positional embeddings to the tokens of each visual domain, where the same positions in different visual domains share identical positional embeddings, providing an explicit cue for the spatial alignment. Besides, we also introduce domain embeddings and role embeddings to help the model distinguish the domains and the roles of the tokens.

## 3.3 DATA CONSTRUCTION

To support joint modeling of multiple visual domains, we require a large-scale dataset containing high-quality images and corresponding labels of various domains. We construct the dataset from two kinds of sources: 1) images with predicted labels and 2) images with ground-truth labels.

First, we collect images with high quality and diversity from several publicly available sources, including Subjects200K (Tan et al., 2024), Aesthetic-4K (Zhang et al., 2025a), and Pexels (opendiffusionai; gaunernst). All of these images have resolutions over $1024\times1024$, which is advantageous for training a high-resolution generative model. As these datasets lack labels, we use state-of-the-art predictors to automatically annotate the data with labels corresponding to 7 specific domains. Specifically, we employ Informative Drawings (Chan et al., 2022) to generate line arts, PiDiNet (Su et al., 2021) to extract edge maps, Depth Anything V2 (Yang et al., 2024) and Lotus (He et al., 2025) to estimate depth maps, Lotus (He et al., 2025) to estimate normal maps, RGB2X (Zeng et al., 2024) to estimate albedos, Oneformer (Jain et al., 2023) to predict segmentation colormaps, and Openpose (Cao et al., 2019) to predict human skeletons. In this manner, we construct a dataset containing 200K images with corresponding $7\times200$K predicted labels, resulting in a total of 1.6M data points. We name this dataset **Joint-1.6M**, and it will be made publicly available.

However, the predicted labels may lack sufficient accuracy, especially for in-the-wild images. To this end, we also employ datasets with ground-truth labels, including BSDS500 (Arbelaez et al., 2010) for edge maps, Hypersim (Roberts et al., 2021) for depth, normal, and albedo maps, and ADE20K (Zhou et al., 2017) for semantic segmentation maps. These datasets encompass a total of 90K images.

Furthermore, we use BLIP2-OPT-2.7b (Li et al., 2023a) and Qwen2-VL-7b-Instruct (Wang et al., 2024a) to generate captions. The former tends to provide a concise description of the main subject in the image, while the latter tends to give a longer paragraph with more details. During the training process, one of these two captions is randomly selected for each image.

## 4 EXPERIMENT

### 4.1 SETUP

**Training Details** We adopt Sana (Xie et al., 2025a) as our base model. We train our model using the CAME-8bit optimizer (Luo et al., 2023) for 130K steps, with a learning rate of $4 \times 10^{-5}$, a batch size of 32, and BF16 mixed-precision, which takes around 535 hours on 8 RTX A6000. Since our dataset contains images with various aspect ratios, we use a ratio bucketing strategy (NovelAI, 2022) during training to prevent important contents from being cropped. This also allows users to generate images with a wide range of aspect ratios during inference.

**Sampling Details** We employ Flow-DPM-Solver (Xie et al., 2025a), a variant of DPM-Solver++ (Lu et al., 2022) adapted for rectified flow. The classifier-free guidance (Ho & Salimans, 2022) scale is set to 4.5. For joint generation and controllable generation, we use 20 sampling steps. For image perception, we use 10 sampling steps since increasing the steps leads to little performance gain.

**Comparison Methods** For unified models, we compare against OmniGen (Xiao et al., 2024), PixWizard (Lin et al., 2025), and OneDiffusion (Le et al., 2025). Their training settings, including the base models, the number of parameters, and the datasets, are listed in Table 1. We also compare against various specialist models on different tasks. For controllable generation, we use ControlNet (Zhang et al., 2023), UniControl (Qin et al., 2024), and EasyControl (Zhang et al., 2025b) as baselines. For geometry (depth and normal) estimation, we include Marigold (Ke et al., 2024), GeoWizard (Fu et al., 2024), GenPercept (Xu et al., 2025a), StableNormal (Ye et al., 2024a), and Lotus (He et al., 2025). For albedo estimation, we compare with Ordinal Shading (Careaga & Aksoy, 2023), Kocsis et al. (2024), Careaga & Aksoy (2024), and RGB2X (Zeng et al., 2024). For edge detection, we adopt HED (Xie & Tu, 2015) and PiDiNet (Su et al., 2021). Additional comparison results are reported in the Appendix.

Table 1: Training settings of unified models.

| Method | Base Model | # Parameters | Dataset |
|---|---|---|---|
| OmniGen | Phi-3 (Abdin et al., 2024) | 3.8B | X2I (100M) |
| PixWizard | Lumina-Next-T2I (Zhuo et al., 2024) | 2B | PixWizard (30M) |
| OneDiffusion | (from scratch) | 2.8B | One-Gen (75M) |
| Jodi (ours) | Sana (Xie et al., 2025a) | 1.6B | Joint-1.6M (200K) + GT labels (90K) |

### 4.2 VISUAL GENERATION AND UNDERSTANDING

**Joint Generation** In Figure 3, we illustrate the capability of our Jodi to simultaneously generate high-quality images of various aspect ratios along with corresponding labels, including depth, normal, albedo, edge, lineart, segmentation, and openpose. The generated images and the generated labels are semantically consistent and spatially aligned, credited to the linear attention and domain-invariant positional embeddings. Please refer to the appendix for more results.

**Controllable Generation** To demonstrate Jodi's performance in controllable generation, we first generate images using existing labels as input conditions and evaluate their fidelity using FID scores (Heusel et al., 2017). Then, to evaluate the faithfulness of the generated images to the input conditions, we re-extract the conditions from the generated images and compare them to the input conditions using LPIPS (Zhang et al., 2018). As shown in Table 2 and Figure 4, Jodi outperforms both existing unified models and generation-only specialist models for all conditions.

Table 2: Quantitative comparison of controllable generation.

| Method | Depth | | Normal | | Edge | | Lineart | | Openpose | |
|---|---|---|---|---|---|---|---|---|---|---|
| | LPIPS↓ | FID↓ | LPIPS↓ | FID↓ | LPIPS↓ | FID↓ | LPIPS↓ | FID↓ | LPIPS↓ | FID↓ |
| ControlNet | 0.29 | 19.5 | 0.35 | 28.0 | 0.23 | 18.9 | 0.33 | 15.9 | 0.11 | 32.0 |
| UniControl | 0.29 | 18.8 | 0.35 | 22.5 | 0.31 | 39.1 | - | - | 0.11 | 26.8 |
| EasyControl | 0.27 | 19.5 | - | - | 0.31 | 20.0 | - | - | 0.12 | 33.9 |
| OmniGen | 0.31 | 20.4 | 0.33 | 24.9 | 0.25 | 23.3 | 0.35 | 102.7 | 0.22 | 33.5 |
| PixWizard | **0.23** | 14.4 | **0.27** | 16.7 | 0.29 | 22.9 | 0.22 | 14.6 | 0.16 | 31.7 |
| OneDiffusion | 0.24 | 15.9 | 0.41 | 21.6 | 0.26 | 40.5 | 0.40 | 37.2 | - | - |
| Jodi (ours) | **0.23** | **13.6** | **0.27** | **13.6** | **0.20** | **13.7** | **0.20** | **11.3** | **0.15** | **23.8** |

*\* First block: specialist models, second block: unified models.*

*\* **Bold**: the best results among unified models.*

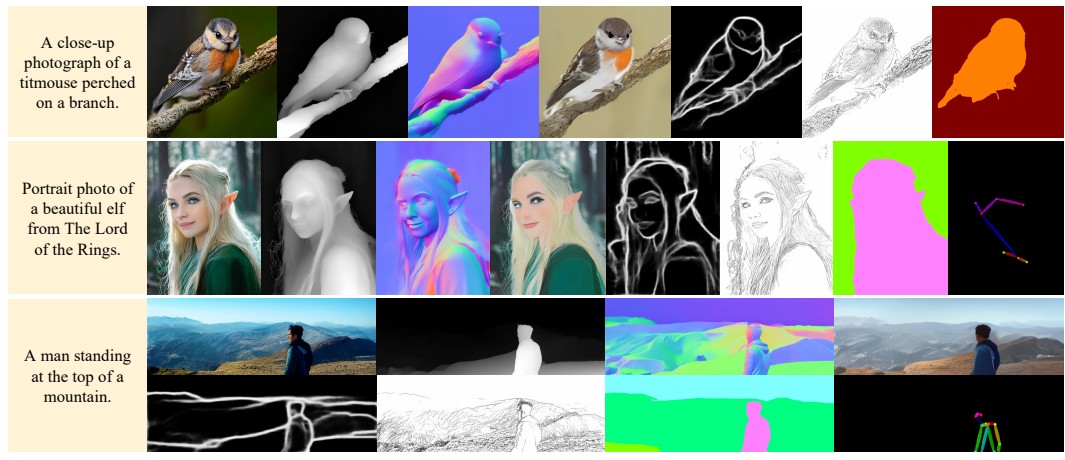

Figure 3: Joint generation of images and labels across a wide range of aspect ratios.

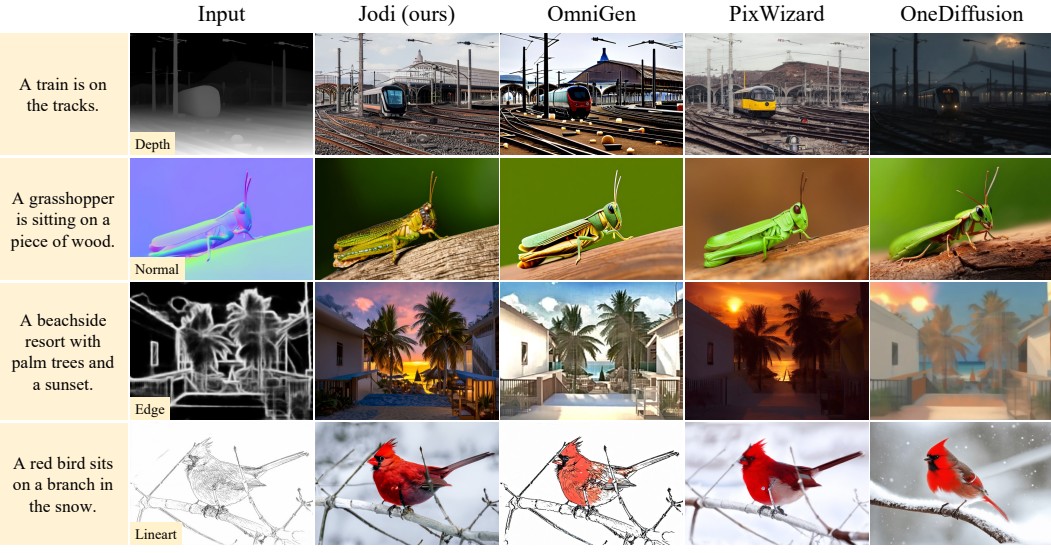

Figure 4: Visual comparison of controllable generation.

**Image Perception** We assess the visual understanding ability of our Jodi on four image perception tasks: depth estimation, normal estimation, albedo estimation, and edge detection. For depth estimation, we evaluate our model on NYUv2 (Silberman et al., 2012), ScanNet (Dai et al., 2017), and DIODE (Vasiljevic et al., 2019) datasets using absolute mean relative error. For normal estimation, we evaluate our model on NYUv2 (Silberman et al., 2012), ScanNet (Dai et al., 2017), and iBims (Koch et al., 2018) datasets using mean angular error. For albedo estimation, we evaluate our model on

the Hypersim (Roberts et al., 2021) test set using PSNR and LPIPS (Zhang et al., 2018). For edge detection, we evaluate our model on the BSDS500 (Arbelaez et al., 2010) test set, using F-scores at Optimal Dataset Scale (ODS) and Optimal Image Scale (OIS) as evaluation metrics. Besides, given the stochastic nature of diffusion models, we also report the ensemble performance by sampling five times for each input image and averaging the results. As shown in Table 3, Table 4, Table 5, Table 6, and Figure 5, our method consistently achieves superior or comparable results to the other unified and specialist models.

Table 3: Quantitative comparison of depth estimation with absolute mean relative error ↓.

| Method | NYUv2 | ScanNet | DIODE |
|---|---|---|---|
| Marigold | 5.5 | 6.4 | 30.8 |
| GeoWizard | 5.6 | 6.4 | 33.5 |
| Lotus-D | 5.1 | 5.5 | 22.8 |
| OmniGen | 9.2 | 10.1 | 30.6 |
| PixWizard | **7.0** | **7.9** | 25.4 |
| OneDiffusion | 8.9 | 9.7 | **25.2** |
| Jodi (ours) | 10.1 | 12.1 | 25.9 |
| Jodi (ours, ensemble) | 8.3 | 9.9 | 25.8 |

*First block: specialist models, second block: unified models.
***Bold**: the best results among unified models.*

Table 4: Quantitative comparison of surface normal estimation with mean angular error ↓.

| Method | NYUv2 | ScanNet | iBims |
|---|---|---|---|
| GeoWizard | 18.9 | 17.4 | 19.3 |
| GenPercept | 18.2 | 17.7 | 18.2 |
| StableNormal | 18.6 | 17.1 | 18.2 |
| Lotus-D | 16.2 | 14.7 | 17.1 |
| OmniGen | 28.9 | 28.9 | 31.3 |
| PixWizard | 23.5 | 26.6 | 22.5 |
| Jodi (ours) | 21.1 | 24.3 | 20.1 |
| Jodi (ours, ensemble) | **18.6** | **20.3** | **18.2** |

*First block: specialist models, second block: unified models.
***Bold**: the best results among unified models.*

Table 5: Quantitative comparison of albedo estimation on the Hypersim test set.

| Method | PSNR↑ | LPIPS↓ |
|---|---|---|
| Ordinal Shading | 15.6 | 0.34 |
| Kocsis et al. (2024) | 11.3 | 0.49 |
| Careaga & Aksoy (2024) | 15.7 | 0.36 |
| RGB2X | 20.6 | 0.18 |
| Jodi (ours) | 15.5 | 0.31 |
| Jodi (ours, ensemble) | 16.5 | 0.33 |

*First block: specialist models, second block: unified models.
***Bold**: the best results among unified models.*

Table 6: Quantitative comparison of edge detection on the BSDS500 test set.

| Method | ODS↑ | OIS↑ |
|---|---|---|
| HED | 0.788 | 0.808 |
| PiDiNet | 0.807 | 0.823 |
| OmniGen | 0.767 | 0.781 |
| PixWizard | 0.605 | 0.633 |
| OneDiffusion | 0.682 | 0.691 |
| Jodi (ours) | **0.826** | **0.851** |

*First block: specialist models, second block: unified models.
***Bold**: the best results among unified models.*

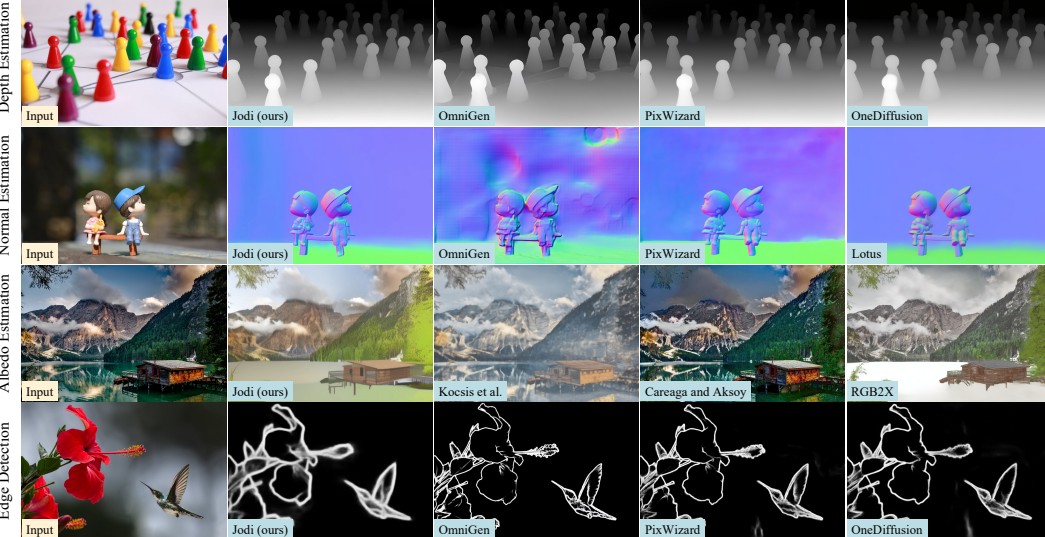

Figure 5: Visual comparison of image perception tasks on in-the-wild images.

### 4.3 ANALYSIS

**Effect of Domain-invariant Positional Embeddings** As described in Section 3.2, we introduce domain-invariant positional embeddings to encourage the spatial alignment across different visual domains. To validate the effect, we compare our models trained for 10K steps with and without positional embeddings, by observing whether the joint generated images and labels are spatially aligned. As shown in Figure 6, our model aligns the image domain and label domains significantly better with positional embeddings, whereas obvious misalignment is observed without positional embeddings.

**Attention Map Visualization** To further investigate how the tokens from different visual domains align and interact with each other, we pick two query tokens from the image domain and visualize the corresponding attention maps in Figure 7. As can be seen, most domains show strong activation at the same spatial location as the query token, demonstrating a good alignment between these domains. Interestingly, attention maps of different domains also reveal their own unique structural patterns. For example, the segmentation domain exhibits strong activation along semantic boundaries, and the openpose domain focuses more on the human figure.

**Joint Consistency** In Figure 8, we illustrate the consistency of our unified model across joint generation, controllable generation, and image perception tasks. We first perform joint generation based on the input prompt to produce samples covering all of the image and label domains. According to each generated label, we then apply controllable generation to generate new images that comply with these labels. Besides, we perform image perception on the image generated in the first step to detect all its labels. As can be observed, three types of tasks produce visually consistent results.

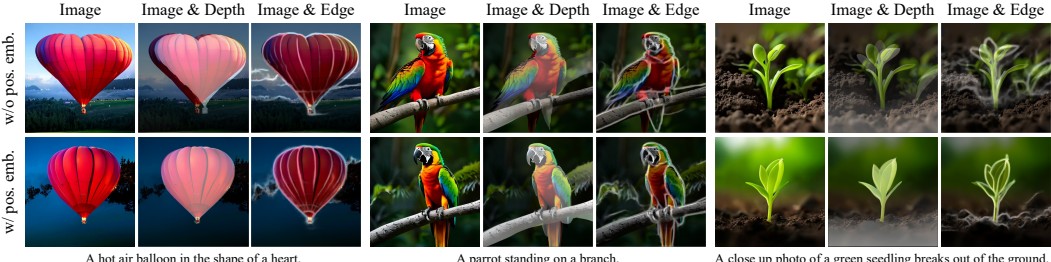

Figure 6: Effect of positional embeddings. Generated labels are overlaid on images for a better view.

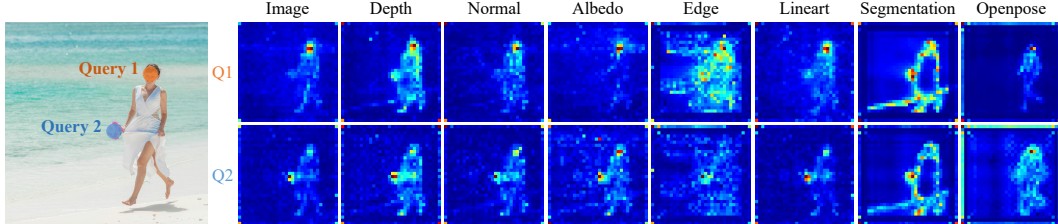

Figure 7: Visualization of attention map.

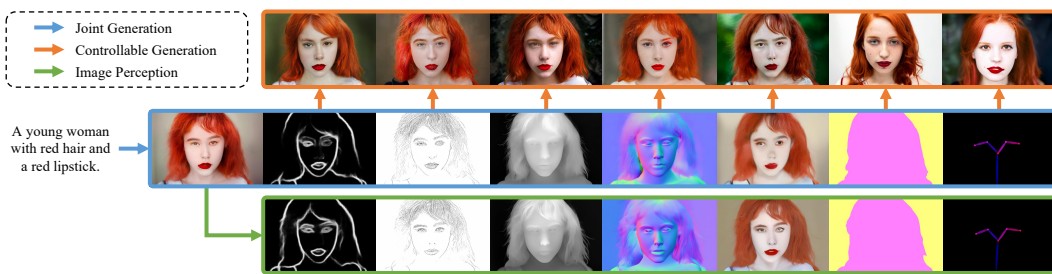

Figure 8: Jodi shows consistency among joint generation, controllable generation, image perception.

**Extension to New Domains** Our well-trained Jodi model can be readily extended to one or more new domains by appending the corresponding tokens to the existing ones. Figure 9 presents the joint generation results after fine-tuning the model on the doodle sketch domain (Arar et al., 2025) as well as simultaneous fine-tuning on the pixel, irradiance, and canny domains.

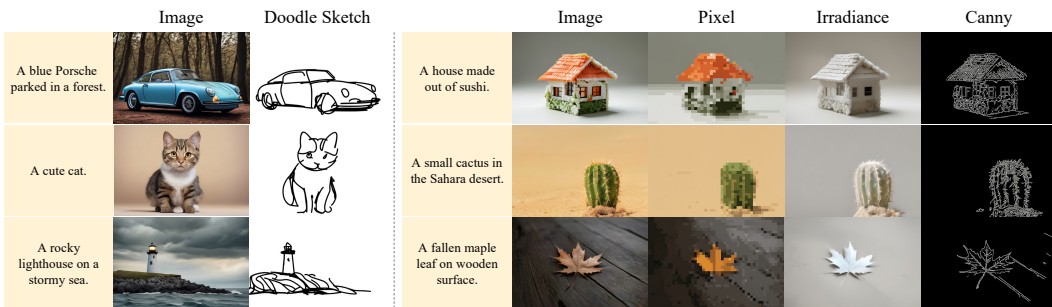

Figure 9: Joint generation results of our model extended to new domains.

## 5 CONCLUSION AND LIMITATIONS

Motivated by the interdependence between generation and understanding inherent in the joint distribution, we propose Jodi, a diffusion framework that jointly models the image domain and multiple label domains, unifying the visual generation and understanding. We design a Role-Switch mechanism that allows the model to simultaneously learn joint generation, controllable generation, and image perception. Furthermore, to facilitate the interaction and alignment between tokens from different visual domains, we introduce masked linear attention and domain-invariant positional embeddings. As a result, our Jodi is capable of both generation and understanding tasks across the image domain and 7 distinct label domains. We also introduce the Joint-1.6M dataset, which will be publicly released to advance this research area.

While Jodi achieves impressive performance, it still comes with certain limitations. First, due to the limited size of our training dataset, the generated images may exhibit structural distortions, especially for human figures. Second, we simply represent each domain in RGB space. As a consequence, our model is currently limited to handling 12 clustered classes for the segmentation domain (see the appendix for details), as increasing the number of classes makes the RGB representations of the segments too similar to be reliably distinguished. Similarly, the RGB space is also not the ideal choice for the openpose domain, where the keypoints are better represented by coordinates. These problems may be resolved by incorporating more data and designing specific encoders and decoders for each visual domain, which we leave for future work.

It is important to note that, as with all generative models, Jodi may inherit biases present in the training dataset and could be misused to generate malicious or unintended content. Users should remain vigilant and comply with the usage policies to mitigate these risks.

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

# APPENDIX

## A   DETAILED ARCHITECTURE

Figure 10 shows the detailed architecture of our Jodi. To incorporate the label domains into our backbone model Sana (Xie et al., 2025a), we add a new patch embedding layer and a new final layer for each label domain. The patch embedding layers are responsible for projecting the encoded tokens to match the input dimension of the backbone model, and the final layers project them back to match the input dimension of the decoder. The patch embedding layers of the label domains are initialized from the pretrained Sana weights of the image domain, while the new final layers are randomly initialized. We find that this initialization strategy leads to the best convergence.

The backbone is a stack of linear transformer blocks, where each block is composed of AdaLN-Zero layers (Peebles & Xie, 2023), a linear attention layer (Katharopoulos et al., 2020), a cross attention layer (Rombach et al., 2022), and a mix FFN layer (Xie et al., 2025a). The scale, shift, and gate parameters of the AdaLN-Zero layers are obtained via an MLP that takes the role embeddings, domain embeddings, and timestep embeddings as input; therefore, these parameters are tailored for each domain, helping the model distinguish the roles and domains.

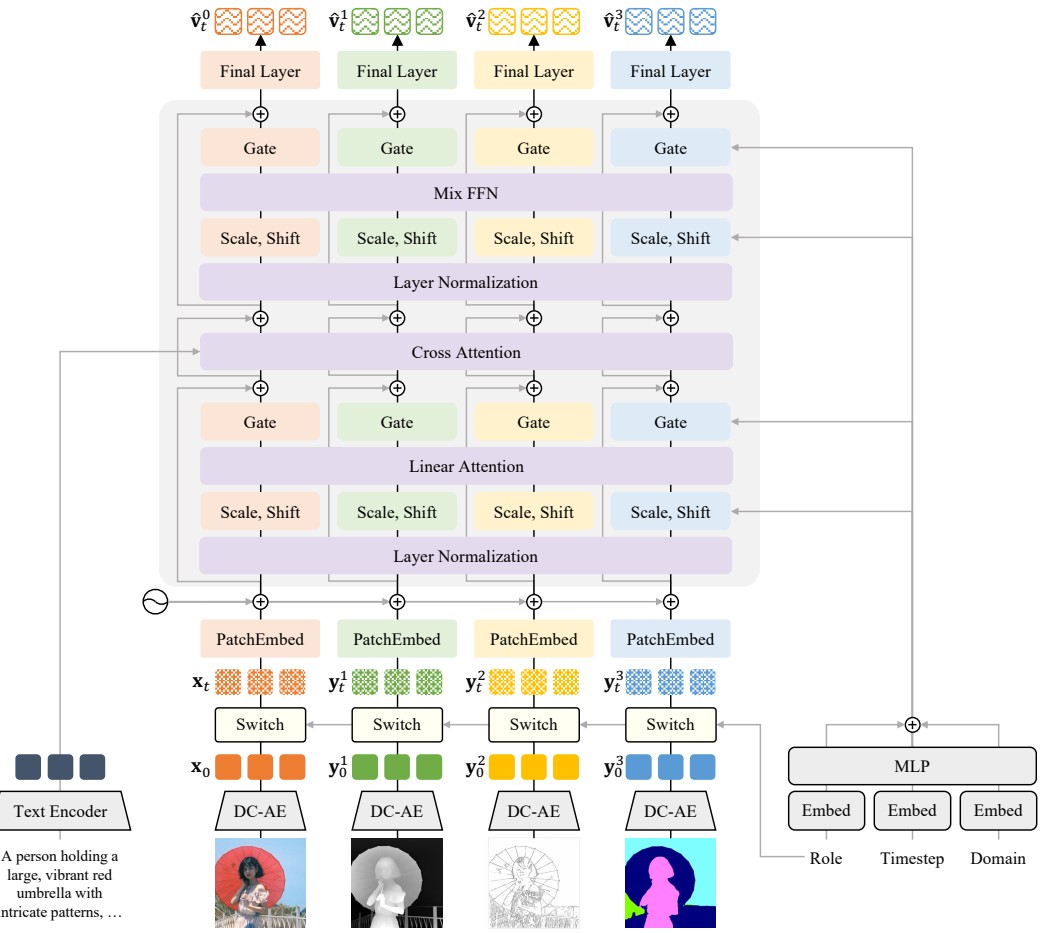

Figure 10: Detailed architecture of Jodi. For clarity, only four domains are illustrated.

## B  RELATIONSHIPS AMONG TEXT, IMAGE, AND LABELS

For joint generation and controllable generation, we can always assume a proper input text describing the content of the image. However, for image perception tasks, the labels are generally determined by the given image alone, regardless of the text description. In the context of graphical models, the labels and the text are conditionally independent given the image, as illustrated by the probabilistic graph in Figure 11. Accordingly, we set the text input to empty for image perception tasks during both training and inference.

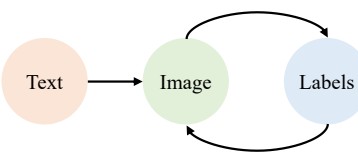

Figure 11: The probabilistic graph of text, image, and labels.

## C  COMPUTATIONAL COST

In Section 3.2, we analyze the theoretical computational complexity of using linear attention versus full attention. In Figure 12, we present the actual VRAM usage, training time, and inference latency when using vanilla full attention (Vaswani et al., 2017), flash attention (Dao et al., 2022), and linear attention (Katharopoulos et al., 2020) in practice. As the number of domains increases, the VRAM usage of vanilla full attention quickly exceeds the memory limits of an RTX A6000 GPU, making our training infeasible. Although flash attention reduces memory usage, its training time is over twice as long as that of linear attention when handling 8 domains, resulting in lower efficiency.

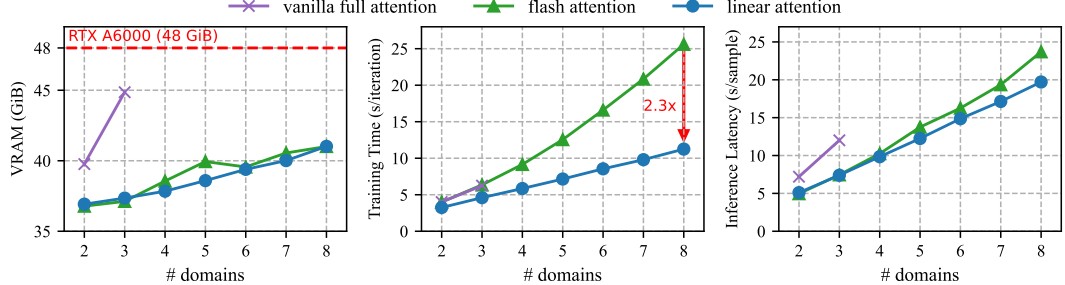

Figure 12: Comparison of actual computational cost among three types of attention.

## D  NOTES ON SEGMENTATION DOMAIN

**Superclasses**   As discussed in the limitation part in Section 5, we represent each visual domain in RGB space, which is not suitable for the semantic segmentation domain. Specifically, the segmentation dataset ADE20K (Zhou et al., 2017) contains as many as 150 semantic classes, where some of the classes are assigned similar or even the same colors in RGB space, causing confusion for the model. To mitigate this problem, we group the 150 classes into 12 manually defined superclasses as shown in Table 7, where the RGB values assigned to different superclasses are set to be as far apart as possible. However, this is apparently not the optimal solution because it decreases the number of distinguishable classes. In the future, we plan to extend our model beyond the RGB space to better accommodate special domains like the segmentation domain.

Table 7: 12 superclasses and the corresponding RGB colors.

| Superclass | Person | Animal | Plant | Water | Mountain | Sky | Building | Vehicle | Wall | Road | Furniture | Others |
|---|---|---|---|---|---|---|---|---|---|---|---|---|
| Color | FF7FFF | FF7F00 | 7FFF00 | 007FFF | 00FF7F | 7FFFFF | FF007F | 7F00FF | FFFF7F | 7F0000 | 007F00 | 00007F |

In Table 8, we report the Intersection-over-Union (IoU) for each superclass except "Others", as well as their mean IoU (mIoU). We also report the ensemble performance by sampling five times for each input image and performing majority voting. For methods trained on the original 150 classes of ADE20K, we map their predictions to our 12 superclasses before computing IoUs. It is worth noting that such a comparison is somewhat unfair to the other methods, because these methods are trained to predict 150 classes, which is a more challenging task than predicting our 12 superclasses.

Table 8: Quantitative comparison on semantic segmentation (12 classes) on ADE20K test set.

| Method | IoU per class | | | | | | | | | | | mIoU |
|--------|--------|--------|-------|-------|----------|------|----------|---------|------|------|-----------|------|
| | Person | Animal | Plant | Water | Mountain | Sky | Building | Vehicle | Wall | Road | Furniture | |
| Uniformer | 78.0 | 62.9 | 75.8 | 64.9 | 61.7 | 93.2 | 87.1 | 76.2 | 87.9 | 74.6 | 81.5 | 76.7 |
| Oneformer | 87.3 | 65.4 | 81.0 | 88.4 | 69.7 | 95.2 | 90.8 | 86.2 | 90.0 | 82.7 | 86.0 | 83.9 |
| PixWizard | 47.1 | 0.0 | 53.0 | 25.4 | 14.4 | 85.1 | 50.3 | 29.7 | 66.1 | 38.9 | 24.5 | 39.5 |
| Jodi (ours) | 74.0 | **14.7** | 55.7 | 50.7 | 37.9 | 90.9 | 67.0 | 52.5 | 72.4 | 61.0 | 56.2 | 57.5 |
| Jodi (ours, ensemble) | **79.5** | 1.9 | **65.6** | **60.9** | **38.9** | **92.4** | **78.0** | **66.4** | **79.4** | **67.5** | **65.4** | **63.3** |

*\* First block: specialist models, second block: unified models.*

*\* **Bold**: the best results among unified models.*

**Color Remapping** Another solution is to remap the 150 classes to new colors, ensuring that they are well distributed across the RGB space. To this end, we divide the RGB space into a $6 \times 6 \times 5$ grid and assign the first 151 colors (including one for the background) to the classes of ADE20K dataset. Please refer to Figure 13 for an intuitive illustration.

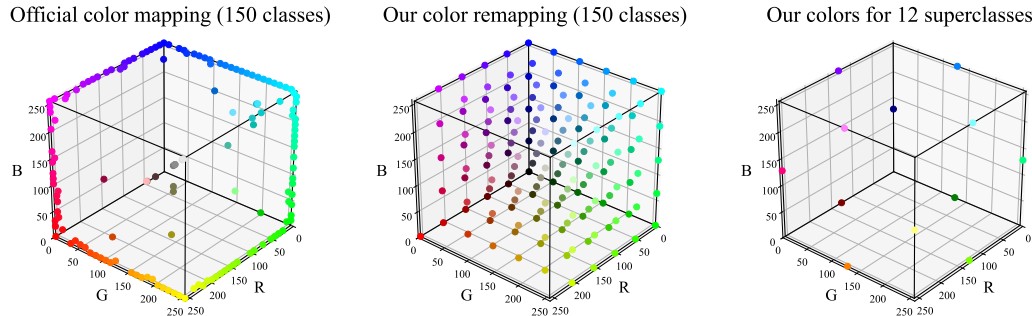

Figure 13: Illustration of different color mappings for ADE20K dataset.

To investigate the performance of our method on the semantic segmentation task with 150 classes, we replace the 12-class segmentation domain in the pretrained Jodi model with the 150-class segmentation domain and fine-tune the model on the ADE20K dataset for 20K steps. The quantitative and qualitative results are shown in Table 9 and Figure 14. Jodi outperforms the previous unified model, PixWizard, even when using the official color mapping. Furthermore, our color remapping strategy significantly improves performance, highlighting the importance of maintaining sufficient separability between classes. However, our results still lag behind those of specialist models. We attribute this gap to the limitations of the RGB space, where distances between colors do not correspond to semantic similarity. For example, semantically related classes "car", "van", and "truck" are mapped to highly distinct colors, while unrelated classes such as "plant" and "sidewalk" are mapped to visually similar colors. This distorted color–semantic relationship introduces unnecessary learning difficulty. In future work, we plan to explore a more suitable representation space beyond RGB for the segmentation domain.

Table 9: Quantitative comparison on semantic segmentation (150 classes) on ADE20K test set.

| Method | mIoU |
|--------|------|
| UniFormer | 44.4 |
| OneFormer | 57.3 |
| PixWizard | 7.0 |
| Jodi (official color mapping) | 11.9 |
| Jodi (our color remapping) | 17.3 |

*\* First block: specialist models, second block: unified models.*

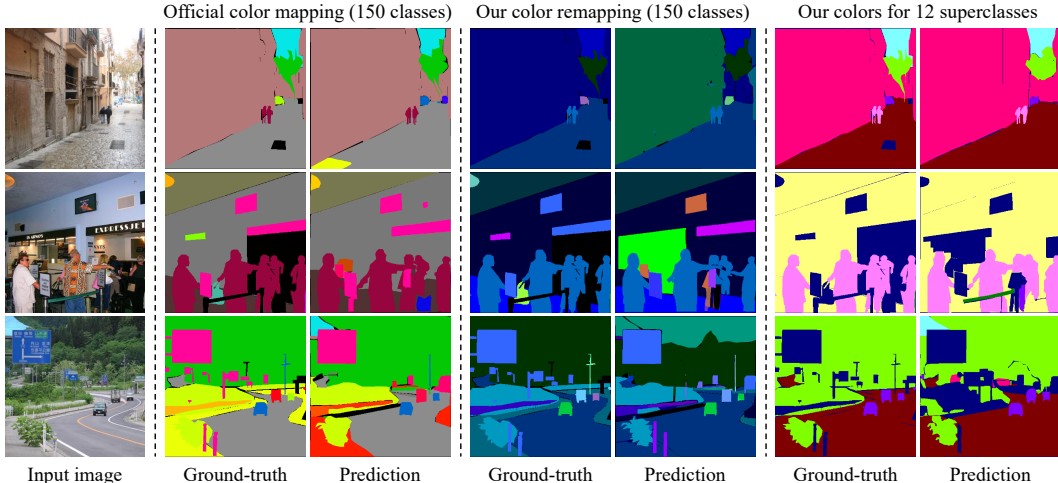

Figure 14: Visualization of semantic segmentation task with three types of color mappings.

## E    REMARKS ON DOMAIN EXTENSION

In Section 4.3, we demonstrate that our model can be efficiently extended to new domains through fine-tuning. Specifically, the fine-tuning process is performed on only one RTX A6000 GPU with a reduced batch size of 4 and a learning rate of $1 \times 10^{-5}$. To illustrate the efficiency, we present the joint generation results throughout the fine-tuning process in Figure 15, comparing with directly training from Sana. It is clear that the newly added domains converge within 2,000 fine-tuning steps (around 10 GPU hours), whereas training from Sana still yields unsatisfactory results even after 5,000 steps. This demonstrates the effectiveness and efficiency of our Jodi in extending new domains.

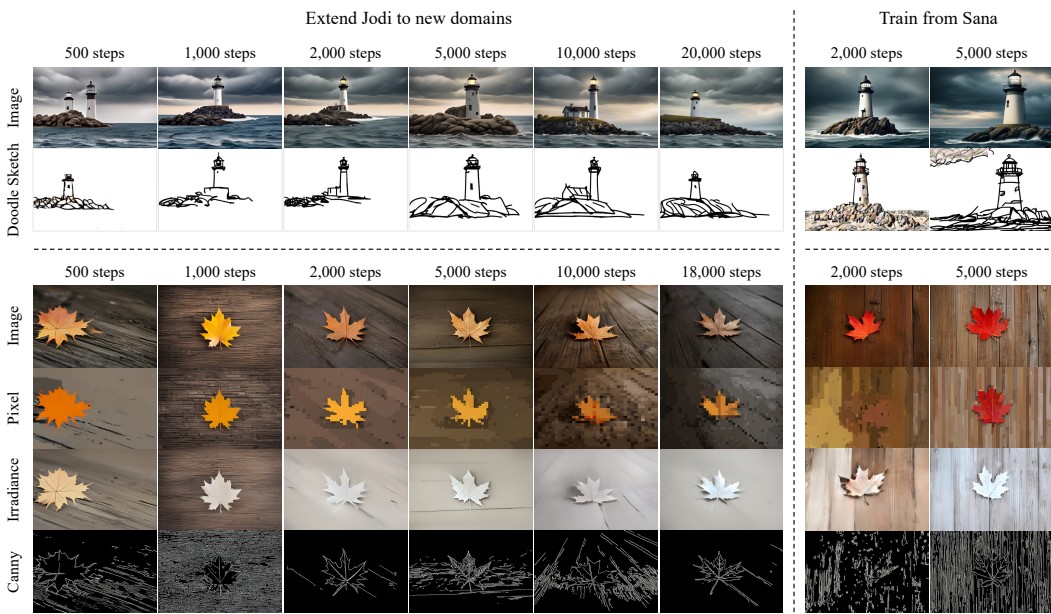

Figure 15: Comparison between extending Jodi to new domains and training from Sana.

# F   ADDITIONAL QUANTITATIVE RESULTS

**Full Results of Depth and Normal Estimation**   We present the full evaluation results of depth estimation in Table 10 and normal estimation in Table 11, where additional methods and metrics are included for a comprehensive comparison. The detailed description of these metrics can be found in Appendix A.2 of the Lotus paper (He et al., 2025).

Table 10: Quantitative comparison on depth estimation.

| Method | NYUv2 | | | ScanNet | | | DIODE | | |
|---|---|---|---|---|---|---|---|---|---|
| | AbsRel$\downarrow$ | $\delta 1\uparrow$ | $\delta 2\uparrow$ | AbsRel$\downarrow$ | $\delta 1\uparrow$ | $\delta 2\uparrow$ | AbsRel$\downarrow$ | $\delta 1\uparrow$ | $\delta 2\uparrow$ |
| Marigold$^\S$ | 5.5 | 96.4 | 99.1 | 6.4 | 95.2 | 98.8 | 30.8 | 77.3 | 88.7 |
| GeoWizard$^\S$ | 5.6 | 96.3 | 99.1 | 6.4 | 95.0 | 98.4 | 33.5 | 72.3 | 86.5 |
| GenPercept$^\S$ | 5.6 | 96.0 | 99.2 | 6.2 | 96.1 | 99.1 | 35.7 | 75.6 | 86.6 |
| Lotus-D$^\S$ | 5.1 | 97.2 | 99.2 | 5.5 | 96.5 | 99.0 | 22.8 | 73.8 | 86.2 |
| OmniGen$^\dagger$ | 9.2 | 91.8 | 98.6 | 10.1 | 90.0 | 98.2 | 30.6 | 71.0 | **85.8** |
| PixWizard$^\dagger$ | **7.0** | **95.0** | **99.1** | **7.9** | **93.7** | **98.8** | 25.4 | 72.1 | 85.0 |
| OneDiffusion$^\dagger$ | 8.9 | 92.0 | 98.2 | 9.7 | 90.7 | 98.0 | **25.2** | **72.2** | 85.3 |
| Jodi (ours) | 10.1 | 89.6 | 97.9 | 12.1 | 84.7 | 96.4 | 25.9 | 69.0 | 84.1 |
| Jodi (ours, w/ ensemble) | 8.3 | 92.7 | 98.8 | 9.9 | 89.4 | 97.8 | 25.8 | 71.0 | 84.9 |

*\* First block: specialist models, second block: unified models.* **Bold** */ underlined: the best / second results among unified models.*
*\*$^\S$ sourced from Lotus (He et al., 2025),  $^\dagger$ evaluated by ourselves following the Lotus protocol.*

Table 11: Quantitative comparison on normal estimation.

| Method | NYUv2 | | | ScanNet | | | iBims | | |
|---|---|---|---|---|---|---|---|---|---|
| | mean$\downarrow$ | 11.25°$\uparrow$ | 30°$\uparrow$ | mean$\downarrow$ | 11.25°$\uparrow$ | 30°$\uparrow$ | mean$\downarrow$ | 11.25°$\uparrow$ | 30°$\uparrow$ |
| GeoWizard$^\S$ | 18.9 | 50.7 | 81.5 | 17.4 | 53.8 | 83.5 | 19.3 | 63.0 | 80.3 |
| GenPercept$^\S$ | 18.2 | 56.3 | 81.4 | 17.7 | 58.3 | 82.7 | 18.2 | 64.0 | 82.0 |
| StableNormal$^\S$ | 18.6 | 53.5 | 81.7 | 17.1 | 57.4 | 84.1 | 18.2 | 65.0 | 82.4 |
| Lotus-D$^\S$ | 16.2 | 59.8 | 83.9 | 14.7 | 64.0 | 86.1 | 17.1 | 66.4 | 83.0 |
| OmniGen$^\dagger$ | 28.9 | 18.1 | 64.5 | 28.9 | 17.7 | 64.7 | 31.3 | 18.3 | 63.1 |
| PixWizard$^\dagger$ | 23.5 | 33.9 | 72.6 | 26.6 | 25.5 | 65.3 | 22.5 | 40.1 | 78.3 |
| Jodi (ours) | 21.1 | 47.7 | 77.7 | 24.3 | 41.3 | 73.9 | 20.1 | 60.0 | 79.6 |
| Jodi (ours, w/ ensemble) | **18.6** | **50.5** | **80.4** | **20.3** | **46.2** | **78.0** | **18.2** | **61.8** | **81.0** |

*\* First block: specialist models, second block: unified models.* **Bold** */ underlined: the best / second results among unified models.*
*\*$^\S$ sourced from Lotus (He et al., 2025),  $^\dagger$ evaluated by ourselves following the Lotus protocol.*

**Multi-conditional Controllable Generation**   In Table 12, we compare our performance of single-conditional and multi-conditional controllable generation. Specifically, we evaluate controllable generation conditioned individually on each of "Depth", "Normal", "Edge", and "Lineart", as well as conditioned on all of them together. Since multiple conditions provide more information than a single condition, it is natural that the former presents better controllability.

Table 12: Comparison between single and multi-conditional controllable generation.

| Method | Depth | | Normal | | Edge | | Lineart | |
|---|---|---|---|---|---|---|---|---|
| | LPIPS$\downarrow$ | FID$\downarrow$ | LPIPS$\downarrow$ | FID$\downarrow$ | LPIPS$\downarrow$ | FID$\downarrow$ | LPIPS$\downarrow$ | FID$\downarrow$ |
| Jodi (single) | 0.23 | 13.6 | 0.27 | 13.6 | 0.20 | 13.7 | 0.20 | 11.3 |
| Jodi (multi) | **0.22** | **10.2** | **0.22** | **10.2** | **0.16** | **10.2** | **0.20** | **10.2** |

**Multi-label Image Perception**   One of the notable features of our Jodi is that it can simultaneously predict multiple types of labels for a given image. In Table 13, we compare the performance of predicting all types of labels at once to predicting one label at a time. As can be seen, the performance of multi-label prediction is slightly inferior to that of single-label prediction, which we attribute to

the absence of ground-truth labels for learning multi-label prediction (we use predicted labels as surrogates). Despite slightly lower performance, predicting all labels at once significantly saves inference time compared to predicting them one by one. For example, performing multi-label prediction 5 times still takes no more inference time than predicting 5 labels individually. Therefore, we can ensemble these 5 repeats of multi-label prediction to achieve better performance, which outperforms single-label prediction in most cases.

Table 13: Comparison between single and multi-label image perception.

| Method | Depth (NYUv2) | | | Normal (NYUv2) | | | Albedo (Hypersim) | | Edge (BSDS500) | | Seg. (ADE20K) |
|---|---|---|---|---|---|---|---|---|---|---|---|
| | AbsRel ↓ | δ1 ↑ | δ2 ↑ | mean ↓ | 11.25° ↑ | 30° ↑ | PSNR ↑ | LPIPS ↓ | ODS ↑ | IDS ↑ | mIoU ↑ |
| Jodi (single) | 10.1 | 89.6 | 97.9 | 21.1 | **47.7** | 77.7 | **15.5** | **0.31** | **0.826** | **0.851** | 57.5 |
| Jodi (multi) | 11.8 | 85.9 | 97.0 | 22.1 | 44.5 | 76.1 | 13.9 | 0.44 | 0.765 | 0.782 | 57.1 |
| Jodi (multi, ensemble) | **9.6** | **90.4** | **98.3** | **19.6** | 46.9 | **79.0** | 15.1 | 0.43 | - | - | **62.2** |

**Effect of Ground-truth Labels** As described in Section 3.3, our dataset is composed of two parts, one with annotated labels and another with ground-truth labels. To demonstrate the necessity of incorporating datasets with ground-truth labels, we compare our Jodi models trained with and without ground-truth labels (both for 20,000 steps). As shown in Table 14, incorporating ground-truth labels significantly improves the performance on all perception tasks.

Table 14: Effect of ground-truth labels.

| Method | Depth (AbsRel ↓) | Normal (mean ↓) | Albedo (PSNR ↑) | Edge (F1-ODS ↑) |
|---|---|---|---|---|
| Jodi | **13.6** | **25.3** | **13.0** | **0.774** |
| Jodi w/o gt labels | 14.7 | 27.1 | 9.2 | 0.756 |

# G  ADDITIONAL VISUAL RESULTS

In this part, we provide additional visual results of our Jodi, including Figure 16 for failure cases, Figure 17 for joint consistency, Figure 18 for joint generation, Figure 19, Figure 20, and Figure 21 for controllable generation, and Figure 22 and Figure 23 for image perception.

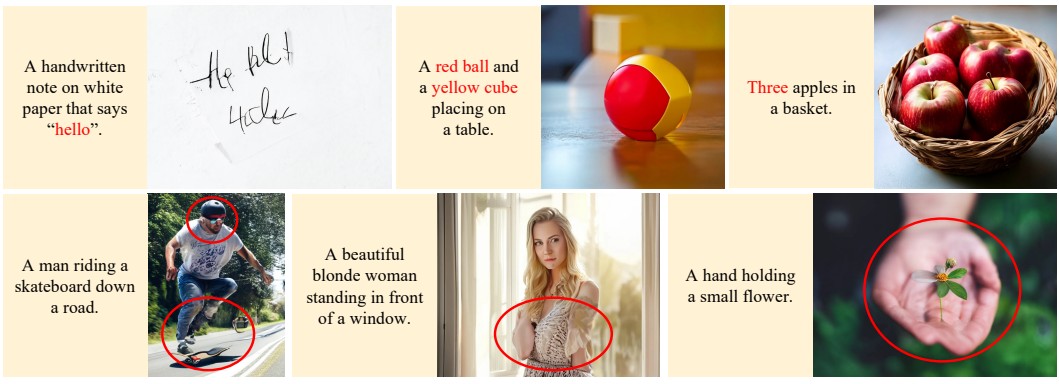

Figure 16: Failure cases of Jodi. The generated images may sometimes exhibit flaws in text rendering, object counting, and structural distortions.

# H  THE USE OF LARGE LANGUAGE MODELS

This paper uses large language models (LLMs) exclusively for language refinement, such as grammar correction and expression improvements. The LLMs are NOT used for generating original ideas, arguments, or research content.

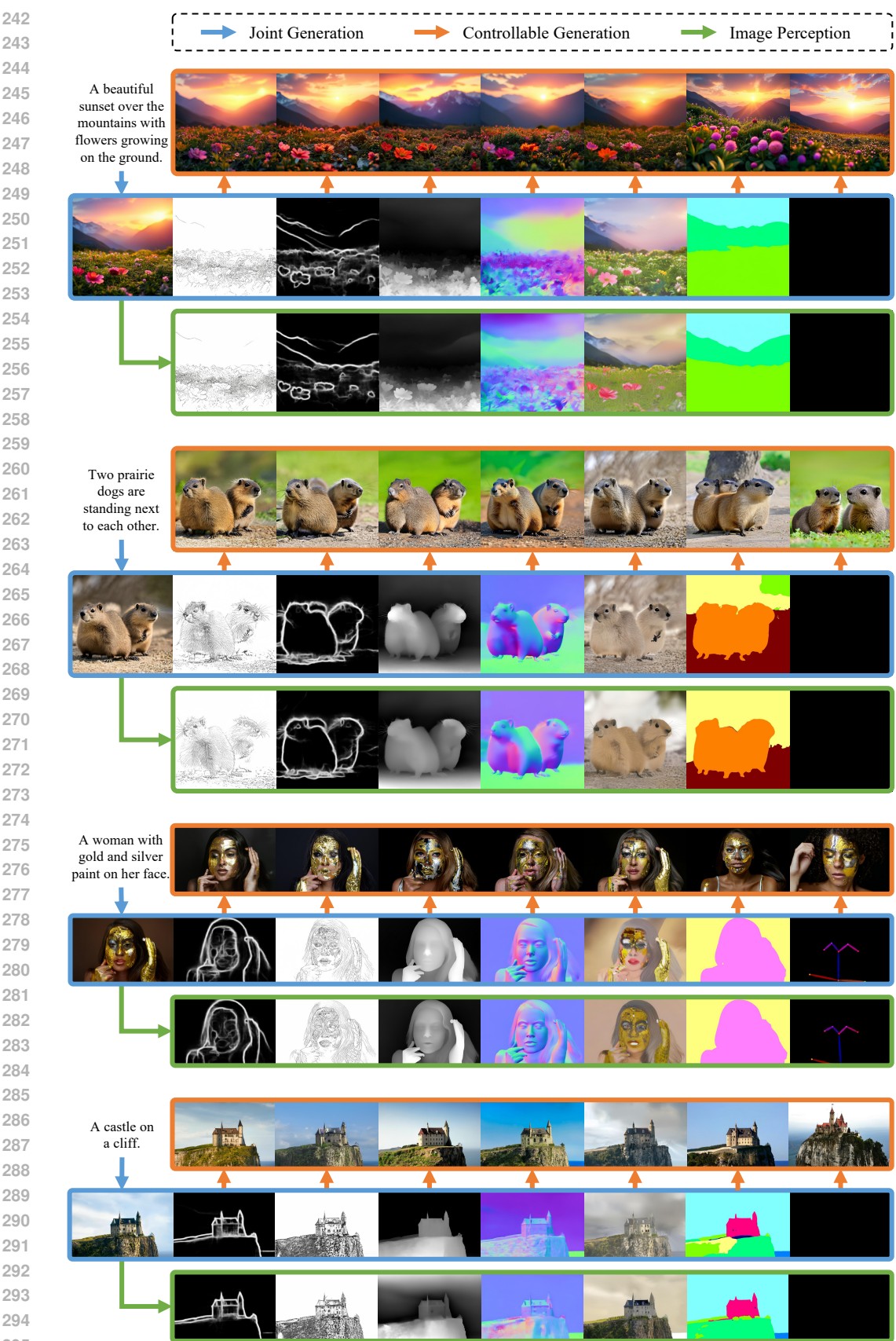

Figure 17: Jodi shows consistency among joint generation, controllable generation, image perception.

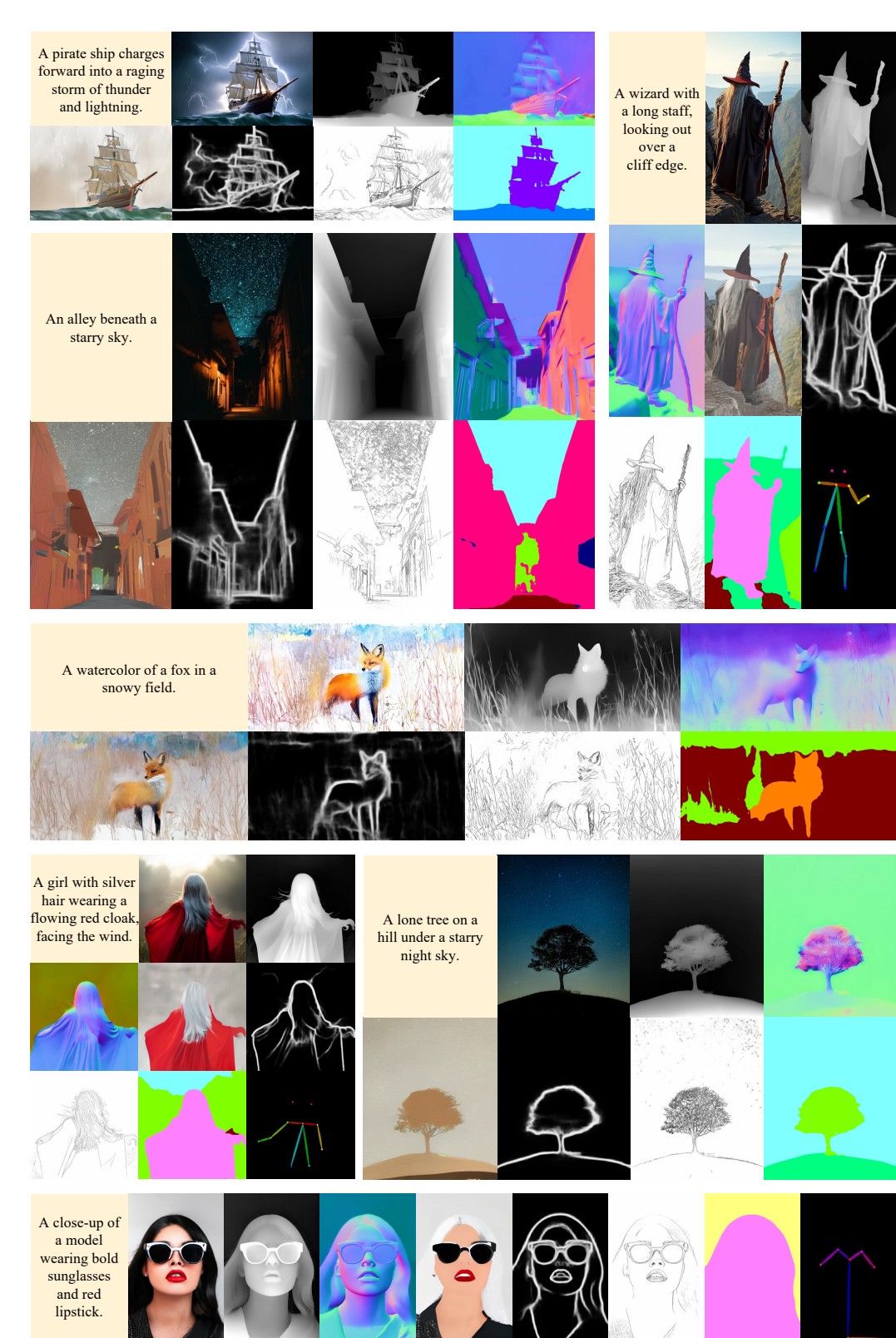

Figure 18: Additional visual results of joint generation.

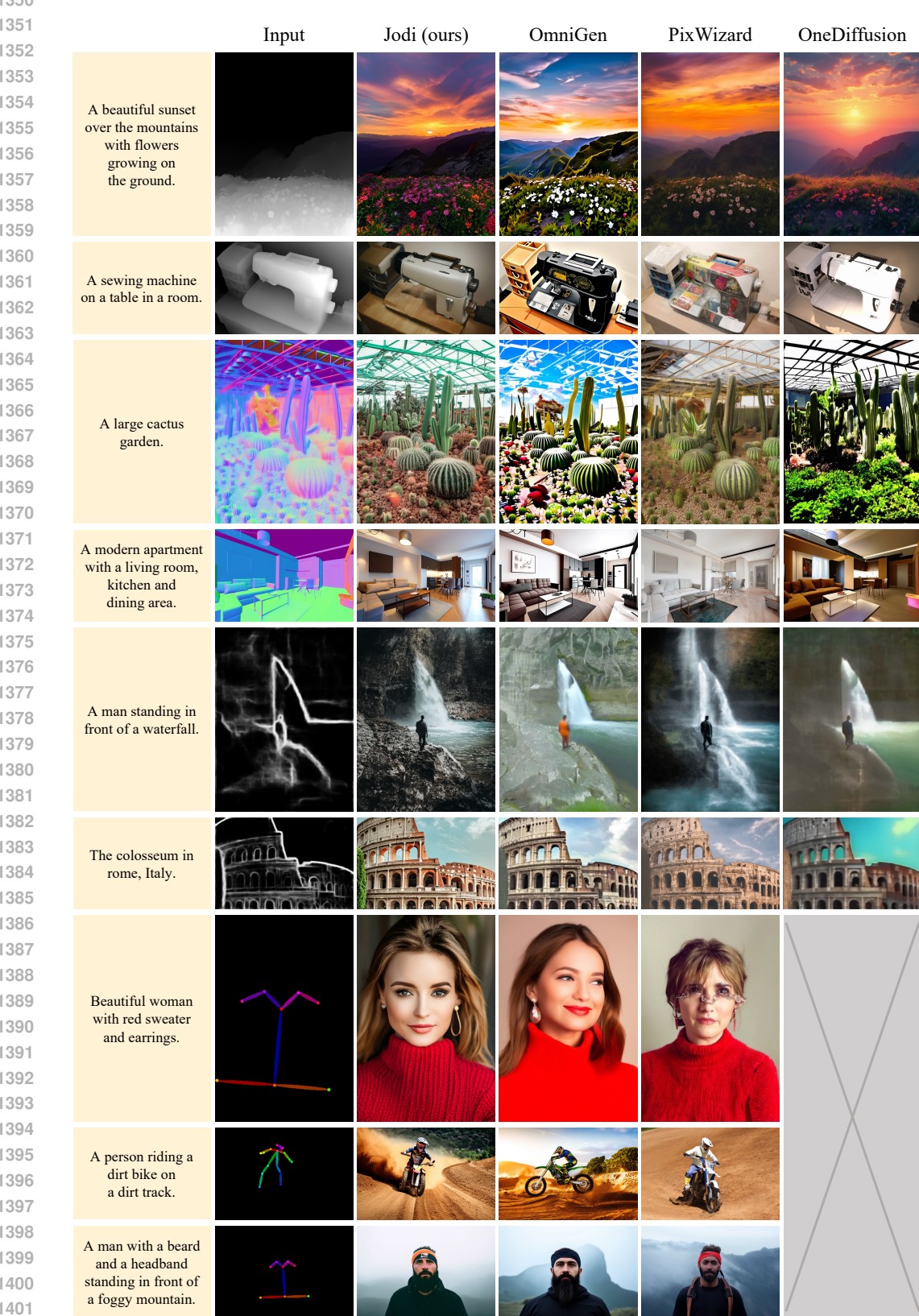

Figure 19: Additional visual comparisons of controllable generation.

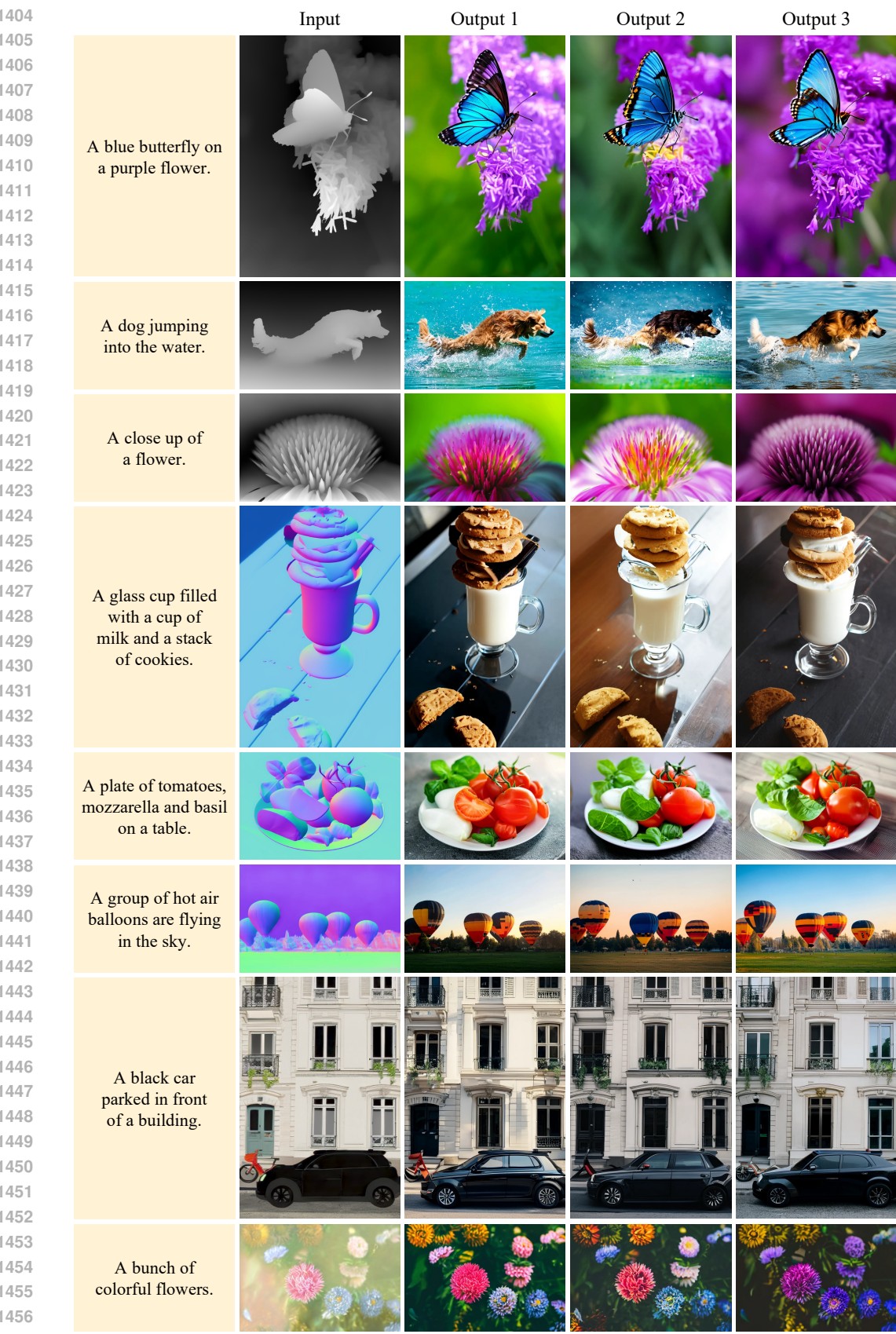

Figure 20: Additional controllable generation results using depth, normal, and albedo inputs.

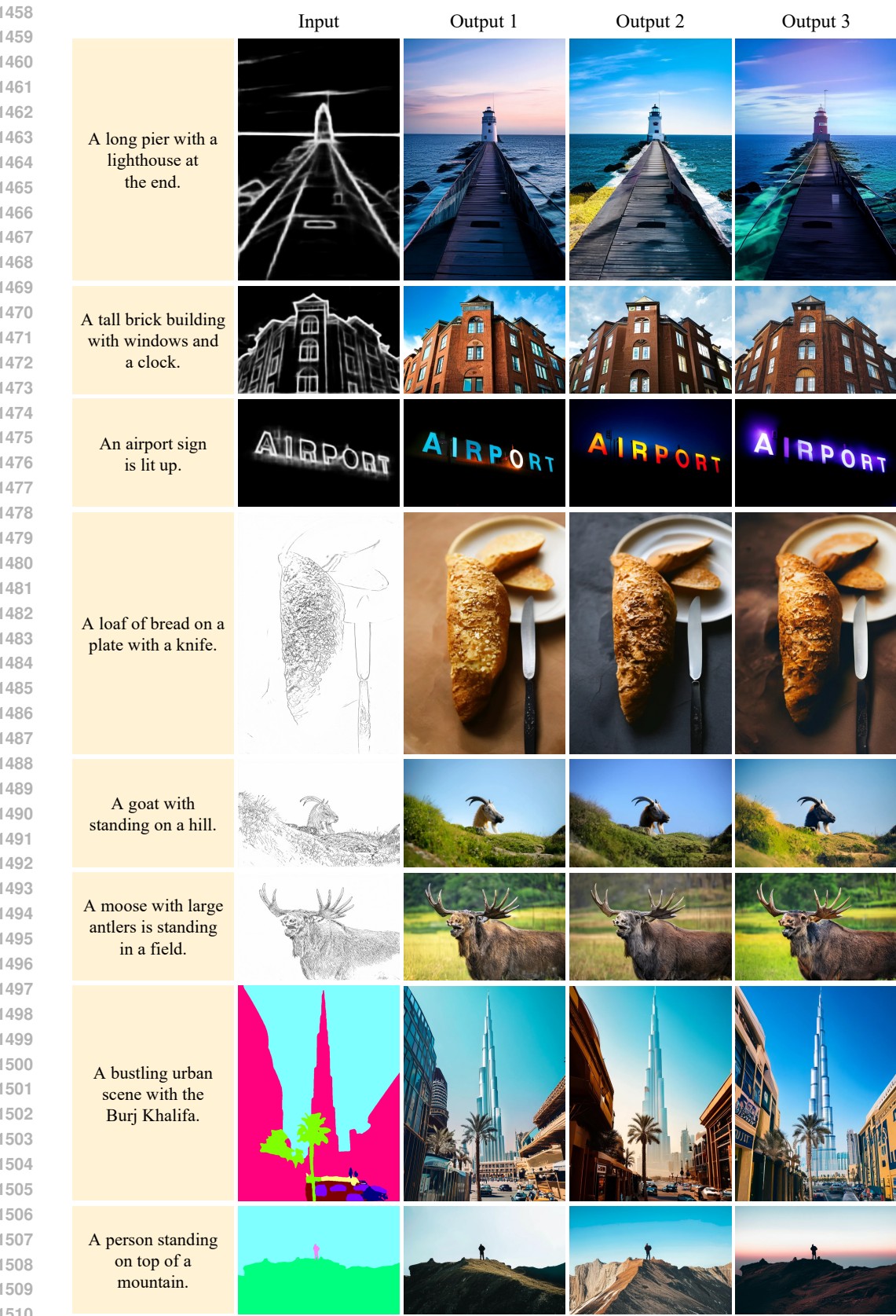

Figure 21: Additional controllable generation results using edge, lineart, and segmentation inputs.

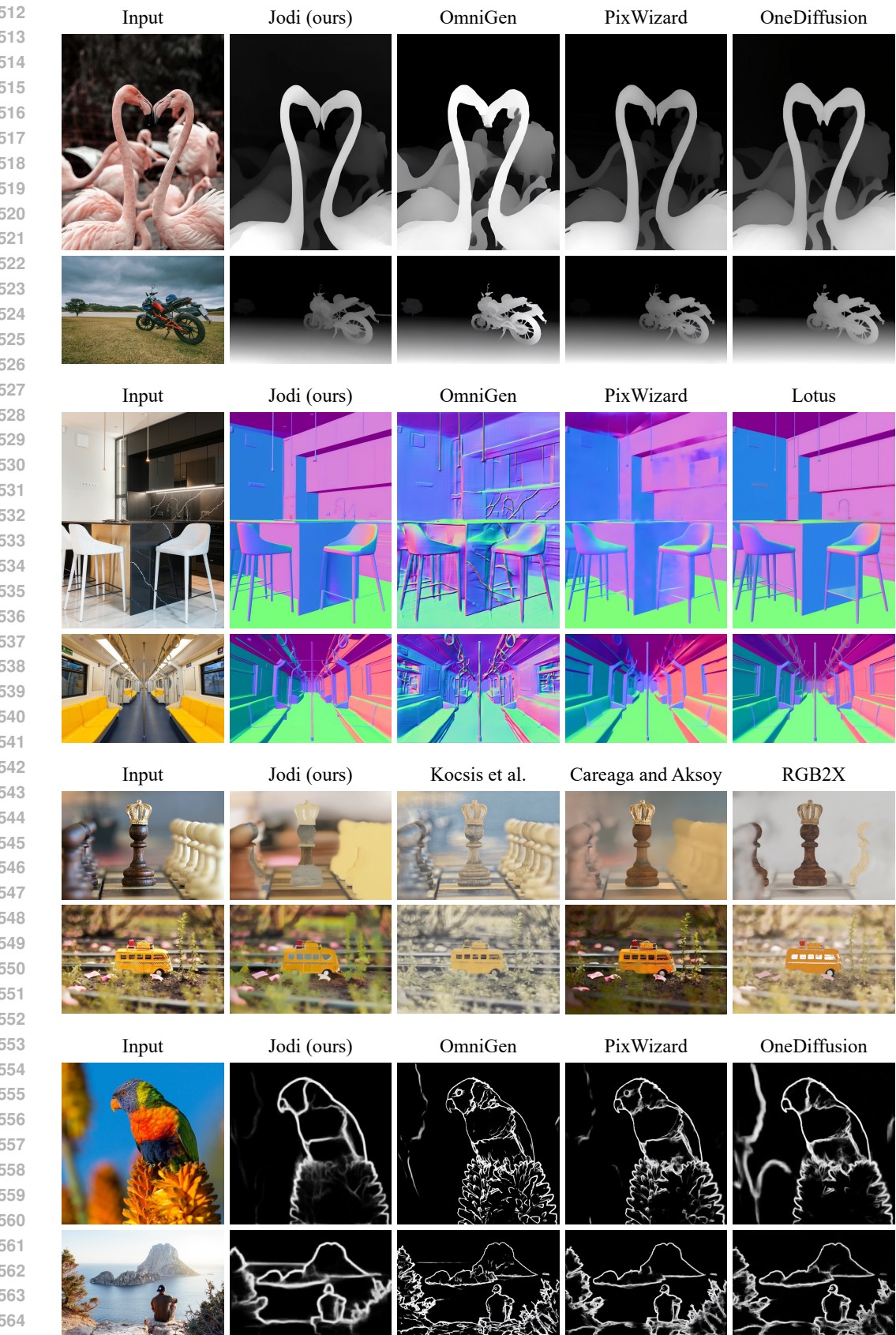

Figure 22: Additional visual comparisons of single-label perception on in-the-wild images.

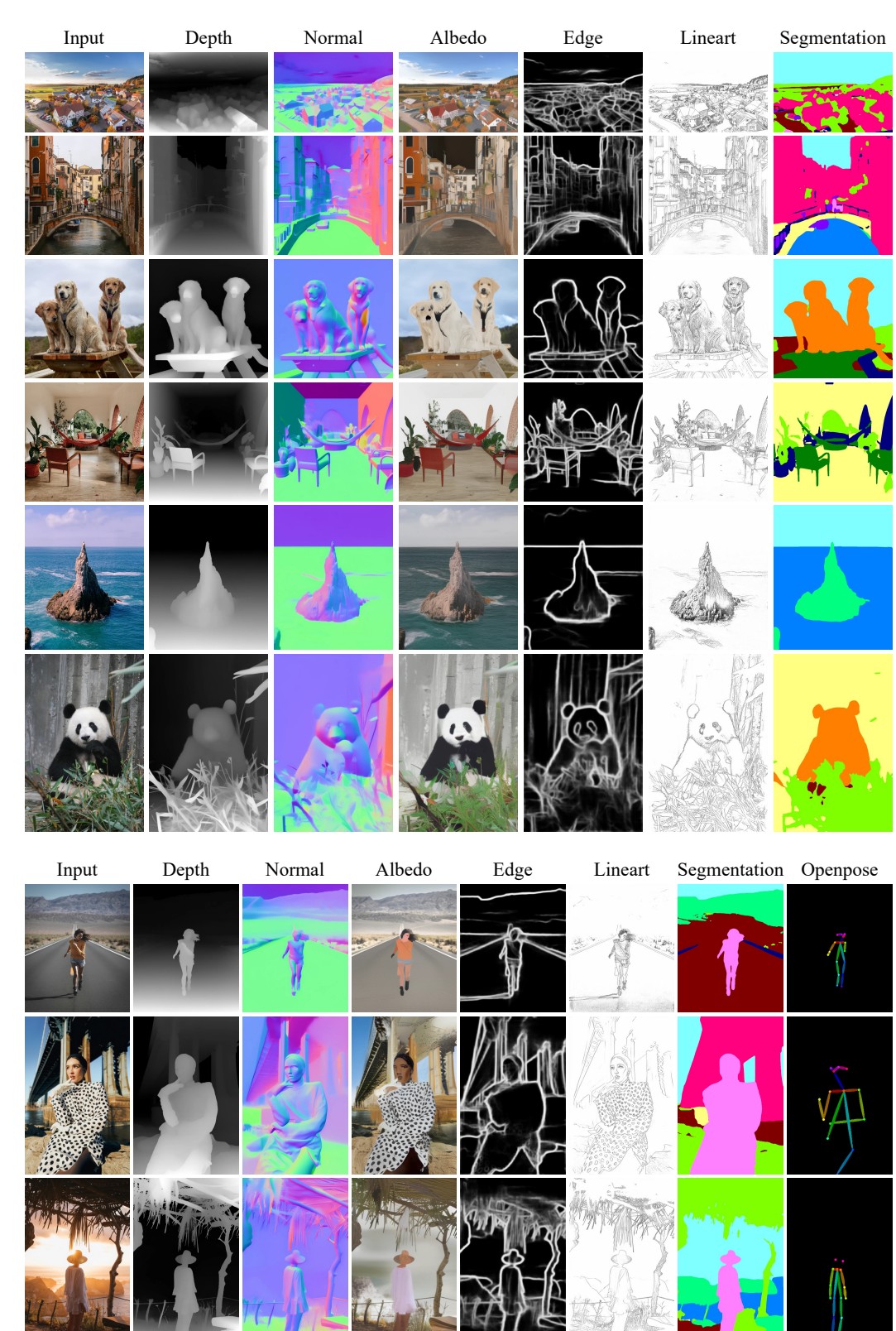

Figure 23: Additional visual results of multi-label perception on in-the-wild images.

# I  LICENSES AND SOURCES

Licenses and sources of datasets and models used in our paper are listed in Table 15 and Table 16.

Table 15: Licenses and sources of datasets used in this paper.

| Dataset | License | Source |
|---|---|---|
| Aesthetic-4K (Zhang et al., 2025a) | MIT | HuggingFace |
| Pexels-photos (opendiffusionai) | Pexels | HuggingFace |
| Pexels-portrait (gaunernst) | Pexels | HuggingFace |
| Subjects200K (Tan et al., 2024) | Apache-2.0 | HuggingFace |
| ADE20K (Zhou et al., 2017) | BSD-3-Clause | Official Website |
| BSDS500 (Arbelaez et al., 2010) | - | Official Website |
| Hypersim (Roberts et al., 2021) | CC BY-SA 3.0 | GitHub |

Table 16: Licenses and sources of models used in this paper.

| Model | License | Source |
|---|---|---|
| Sana-1600M-1024px-BF16 (Xie et al., 2025a) | NVIDIA | GitHub |
| BLIP2-OPT-2.7b (Li et al., 2023a) | MIT | GitHub |
| Qwen2-VL-7b-Instruct (Wang et al., 2024a) | Apache-2.0 | GitHub |
| Depth Anything V2 (Yang et al., 2024) | CC BY-NC 4.0 | GitHub |
| Informative Drawings (Chan et al., 2022) | MIT | GitHub |
| Lotus (He et al., 2025) | Apache-2.0 | GitHub |
| Oneformer (Jain et al., 2023) | MIT | GitHub |
| Openpose (Cao et al., 2019) | Openpose | GitHub |
| PiDiNet (Su et al., 2021) | PiDiNet | GitHub |
| RGB2X (Zeng et al., 2024) | Adobe | GitHub |

