# OpenReview forum: "Jodi: Unification of Visual Generation and Understanding via Joint Modeling"
_ICLR.cc/2026/Conference — Submitted to ICLR 2026_

### Official Review · Reviewer_Qnvb · 2025-10-23

**Soundness:** 2
**Presentation:** 3
**Contribution:** 2
**Rating:** 2
**Confidence:** 3

**Summary:**

This paper proposes a joint modeling framework to train an all-in-one generative model that can do joint generation of images and caption, conditional image generation, and label prediction.

**Strengths:**

- I like the probabilistic perspective on joint modeling p(x, y) = p(x | y) p(y) = p(y | x) p(x). Although simple, it provides an intuitive motivation for why you would want to pursue such a method.
- Visual results clearly show that your method works.
- The paper is well-written and easy to follow.

**Weaknesses:**

see questions

**Questions:**

- Why would somebody want to use this instead of just chaining a few off-the-shelf models to do the target tasks? I get that there's some elegance to having it all in one model, but the pretrained models are already so powerful, that I'm skeptical that somebody would use this approach in practice. Furthermore, the results show that the specialist models often beat Jodi (although Jodi is better than the other joint models).
- What is the methodological novelty?
- Have you considered more modern fast attention mechanisms? [1] Or why not only train a few conditions at a time to get around the quadratic scaling?

[1] Guo, Han, et al. "Log-linear attention." arXiv preprint arXiv:2506.04761 (2025).

---

> ### Author Response · Authors · 2025-11-24
> **Response to Reviewer Qnvb (Part 1/2)**
>
> We highly appreciate your comments and feedback.
>
> &nbsp;
>
> > **Q1**: Why would somebody want to use this instead of just chaining a few off-the-shelf models to do the target tasks? I get that there's some elegance to having it all in one model, but the pretrained models are already so powerful, that I'm skeptical that somebody would use this approach in practice. Furthermore, the results show that the specialist models often beat Jodi (although Jodi is better than the other joint models).
>
> Although today's specialist models are powerful, we see great potential behind the unified models and believe that they can surpass specialist models with further development in the future.
>
> **Evidence from the Evolution of LLMs**
>
> Closely related evidence comes from the development of LLMs, which unify diverse generation and understanding tasks in NLP. Looking back at the early stages of LLMs, such as the representative GPT-3 [1], it was often beaten by specialist models on many tasks, such as ARC Challenge and Winogrande. Nonetheless, subsequent developments, such as GPT-4 [2], demonstrated that LLMs can significantly surpass specialist models. Today, modern LLMs such as GPT-5 [3] are widely used in our daily lives, bringing unprecedented convenience and efficiency to a wide variety of tasks. This evolution trajectory of LLMs indicates that even if unified models initially lag behind specialist models, they have great potential to outperform specialist models and become widely adopted in practical applications.
>
> **Strong Potential of Our Jodi**
>
> In this paper, we propose to unify various visual generation and visual understanding tasks within a single model, under the novel idea of "joint modeling". Notably, for controllable generation tasks, our Jodi already performs better than or on par with specialist models (Table 2). Besides, for perception tasks such as normal estimation and edge detection, Jodi surpasses other unified models and reaches the performance of specialist models (Table 4 & 6). Based on these results, although it currently lags behind specialist models in some other perception tasks, we believe that Jodi has strong potential for continuous improvement and development, just as seen in the evolution of LLMs.
>
> **Practical Value of Our Jodi**
>
> From a practical standpoint, Jodi brings great convenience in deployment. Users no longer need to manage multiple environments, download separate model weights, and run different inference pipelines for each task. As a simple check, we list the storage requirements for each specialist model in the table below. Completing these tasks with specialist models requires a total of 24.5 GiB of storage, whereas Jodi requires only 6.1GiB. In addition, Jodi can predict multiple labels simultaneously in a single sampling process, while using specialist models requires a sequence of separate inference processes. Furthermore, Jodi can perform joint generation, which provides practical value for data synthesis and other downstream applications, as noted by Reviewer yWSM: "The proposed Joint generation could be very valuable in many downstream applications, such as artistic manipulation or asset usage in the traditional frameworks."
>
> | Task | Model | Storage |
> |---|---|---|
> | Depth Estimation | Lotus-D | 4.1GiB |
> | Normal Estimation | Lotus-D | 4.1GiB |
> | Albedo Estimation | RGB2X | 4.9GiB |
> | Edge Detection | PiDiNet | 2.7MiB |
> | Segmentation | OneFormer &nbsp; | 906MiB |
> | Controllable Generation (5 conditions) &nbsp; | ControlNet | 4.0GiB (SD1.5) + 1.3GiB x 5 |
> | Total | | 24.5GiB |
> | Jodi (ours) | | 6.1GiB |
>
> In summary, building on the novel idea of "joint modeling" for unifying visual generation and understanding, Jodi shows both strong potential for future development and practical value for deployment convenience. We believe it represents a meaningful step toward unified vision models.

---

> ### Author Response · Authors · 2025-11-24
> **Response to Reviewer Qnvb (Part 2/2)**
>
> > **Q2**: What is the methodological novelty?
>
> The core contribution of our work is introducing a probabilistic perspective for unifying various visual generation and understanding tasks (i.e., $p(x,y)=p(x|y)p(y)=p(y|x)p(x)$), which is also recognized as a strength in your review. Building on this perspective, we design a novel Role-Switch mechanism to cover diverse conditional distributions within one model, which is recognized as a strength by Reviewer yWSM and LQZ6. In addition, we adopt the masked linear attention to reduce computational complexity and propose domain-invariant positional embeddings for better cross-domain spatial alignment. Together, these techniques contribute to the methodological novelty of our paper.
>
> &nbsp;
>
> > **Q3.1**: Have you considered more modern fast attention mechanisms?
>
> We did not use log-linear attention in this work mainly because it was originally proposed for language modeling and has not yet been validated for image generation tasks. Moreover, according to their paper [4], linear attention is more efficient than log-linear attention during training ($\mathcal{O}(T)$ vs. $\mathcal{O}(T\log T)$), which is helpful for accelerating the development of our framework. Nevertheless, since log-linear attention is indeed theoretically more expressive than linear attention, we will consider incorporating it into our framework in future work. We appreciate your constructive suggestion.
>
> &nbsp;
>
> > **Q3.2**: Why not only train a few conditions at a time to get around the quadratic scaling?
>
> For the joint generation task, where the model simultaneously generates both image and label domains, and multi-label perception task, where the model simultaneously predicts multiple labels from an input image, all domains/conditions are needed during training.
>
> &nbsp;
>
> ---
>
> **We once again thank Reviewer Qnvb for your valuable feedback. We sincerely hope you can re-evaluate our contributions based on our responses.**
>
> &nbsp;
>
> [1] Brown, Tom, et al. "Language models are few-shot learners." Advances in neural information processing systems 33 (2020): 1877-1901.
>
> [2] Achiam, Josh, et al. "Gpt-4 technical report." arXiv preprint arXiv:2303.08774 (2023).
>
> [3] OpenAI. "Introducing GPT-5." https://openai.com/index/introducing-gpt-5/
>
> [4] Guo, Han, et al. "Log-linear attention." arXiv preprint arXiv:2506.04761 (2025).

---

> > ### Comment · Reviewer_Qnvb · 2025-11-25
> > **Recieved**
> >
> > Thanks for the detailed rebuttal. It does make sense, so I’ve raised my score.

---

### Official Review · Reviewer_qawp · 2025-10-26

**Soundness:** 3
**Presentation:** 3
**Contribution:** 3
**Rating:** 6
**Confidence:** 3

**Summary:**

The paper proposes Jodi, a unified diffusion model that can jointly generate images and their corresponding conditions (depth, mask, etc). To achieve this, Jodi considers a unified input format (role switching) and utilizes domain-invariant positional embeddings to optimize flow-based DiT model. Extensive experiments on various benchmarks demonstrate their methods efficacy in unified generation.

**Strengths:**

I like this paper due to the following strengths:

(1) Flexible Control: The framework naturally supports complex, multi-modal conditioning, offering unparalleled flexibility for creative applications.

(2) Enhanced Generalization: By forcing the model to simultaneously learn the generative process  and the analytical structure, JODI is likely to develop a richer, more robust latent representation of visual concepts.

**Weaknesses:**

(1) Lack of T2I evaluation. As demonstrated in Fig. 11, the generated image is the bridge between text and other condisions. Therefore, I would like to see some comparison with T2I methods on GenEval or other T2I benchmarks.

(2) The proposed method achieves condition to image by modeling the joint distribution of various conditions. However, some generation tasks (e.g., depth to image [a]) could be evaluated in a more precise manner. The authors should compare, or at least, discuss with such related works.

(3) Minor: In Tab. 1-openpose, controlnet seems to achieve better result than yours in terms of LPIPS.


[a] 3dis: Depth-driven decoupled instance synthesis for text-to-image generation. In ICLR'25.
[b] 3DIS-FLUX: simple and efficient multi-instance generation with DiT rendering. In Arxiv.

**Questions:**

Please refer to "weaknesses"

---

> ### Author Response · Authors · 2025-11-24
> **Response to Reviewer qawp**
>
> We sincerely thank Reviewer qawp for the comments and positive feedback.
>
> &nbsp;
>
> > **W1**: Lack of T2I evaluation. As demonstrated in Fig. 11, the generated image is the bridge between text and other conditions. Therefore, I would like to see some comparison with T2I methods on GenEval or other T2I benchmarks.
>
> The T2I evaluation results are shown in the table below. Since we mainly focus on the unification of generation and understanding tasks, T2I task is not prioritized over other tasks. Consequently, image quality remains one of our main limitations, which is also noted in Section 5 of the paper. We did not specifically curate data for complex scenes, such as images with multiple objects or human subjects. Also, we did not apply high-quality fine-tuning or preference optimization techniques that are commonly used in T2I models. We believe that this issue could be alleviated by collecting more high quality data and applying advanced techniques, which we leave for future improvement.
>
> | Method | Params. (B) | GenEval ↑ | ImageReward ↑ | HPSv2 ↑ |
> |---|---|---|---|---|
> | SDXL | 2.6 | 0.55 | 0.69 | 27.7 |
> | SD3-medium | 2.0 | 0.62 | 0.86 | - |
> | FLUX-schnell | 12.0 | 0.71 | 0.91 | - |
> | Jodi | 1.6 | 0.54 | 0.58 | 28.0 |
>
> &nbsp;
>
> > **W2**: The proposed method achieves condition to image by modeling the joint distribution of various conditions. However, some generation tasks (e.g., depth to image [a]) could be evaluated in a more precise manner. The authors should compare, or at least, discuss with such related works.
>
> Thank you for bringing the excellent 3DIS [1] and 3DIS-FLUX [2] to our attention. We have added relevant discussion and citations in the Related Work section of the revised paper, highlighted in red.
>
> &nbsp;
>
> > **W3**: Minor: In Tab. 1-openpose, controlnet seems to achieve better result than yours in terms of LPIPS.
>
> In all tables in our paper, we highlight the best results among unified models in bold, excluding specialist models. We have clarified the meaning of the bold numbers in the revised paper.
>
> &nbsp;
>
> [1] Zhou, Dewei, et al. "3dis: Depth-driven decoupled instance synthesis for text-to-image generation." arXiv preprint arXiv:2410.12669 (2024).
>
> [2] Zhou, Dewei, et al. "3dis-flux: simple and efficient multi-instance generation with dit rendering." arXiv preprint arXiv:2501.05131 (2025).

---

### Official Review · Reviewer_LQZ6 · 2025-11-01

**Soundness:** 3
**Presentation:** 3
**Contribution:** 2
**Rating:** 4
**Confidence:** 3

**Summary:**

This paper presents Jodi, a unified diffusion framework designed to jointly model the distribution of images ($x$) and multiple label domains ($y^n$), such as depth, normal, and segmentation. The goal is to unify visual generation and understanding, which are typically treated as separate tasks. The core technical contribution is a "Role-Switch" mechanism, where each domain is randomly assigned as a generation target [G], a condition [C], or ignored [X] during training. This principled approach, based on joint probability modeling, allows the single model to perform three distinct tasks: joint generation ($p(x, y, ...)$), controllable generation ($p(x|y, ...)$), and multi-label image perception ($p(y, ...|x)$). The framework is built on an efficient linear diffusion transformer to handle the computational load of many domains and is trained on a newly curated "Joint-1.6M" dataset.

**Strengths:**

1. Principled and Elegant Framework: The core idea of unifying $p(x|y)$ and $p(y|x)$ by modeling the joint distribution $p(x, y)$ is statistically elegant. The "Role-Switch" mechanism is a clever and direct implementation of this principle, forcing the model to learn a wide range of conditional and marginal distributions within one architecture.
2. The authors made smart architectural choices. The use of a linear diffusion transformer (Sana) correctly identifies and solves the $\mathcal{O}(M^2)$ computational bottleneck of multi-domain ($M$) attention. The "masked linear attention" and "domain-invariant positional embeddings" are solid supporting contributions that are well-motivated and essential for the framework to function.
3. The paper introduces the "Joint-1.6M" dataset (200K images + 7 predicted labels) and aggregates 90K images with ground-truth labels. This is a valuable contribution to the community that will enable future research in joint visual modeling.

**Weaknesses:**

The model's "unification" comes at a significant cost to specialist performance. Though perform better than omni-models on edge detection and normal estimation tasks, Jodi performs noticeably worse than SOTA specialist models on albedo estimation and depth estimation. For example, in depth estimation (Table 2), Jodi achieves 10.1 AbsRel on NYUv2, while the specialist Lotus-D achieves 5.1. In albedo estimation (Table 4), Jodi gets 15.5 PSNR, while the specialist RGB2X gets 20.6. The model excels at generality but does not "excel in... understanding tasks" as claimed.

**Questions:**

1. The perception results (especially Tables 2, 4) are lower than some SOTA unified models and specialist models significantly. Is this performance gap an unavoidable cost of the "jack of all trades" unification, or do the authors see a clear path for Jodi to actually surpass some specialist models in understanding tasks?
2. What was the sampling strategy for the [G], [C], and [X] roles? Were they sampled uniformly at random for all domains? Or was a curriculum used (e.g., more [C] roles early in training) to stabilize the learning of this complex joint distribution?

---

> ### Author Response · Authors · 2025-11-24
> **Response to Reviewer LQZ6**
>
> We highly appreciate your valuable review.
>
> &nbsp;
>
> > **W**: The model's "unification" comes at a significant cost to specialist performance. Though perform better than omni-models on edge detection and normal estimation tasks, Jodi performs noticeably worse than SOTA specialist models on albedo estimation and depth estimation. For example, in depth estimation (Table 2), Jodi achieves 10.1 AbsRel on NYUv2, while the specialist Lotus-D achieves 5.1. In albedo estimation (Table 4), Jodi gets 15.5 PSNR, while the specialist RGB2X gets 20.6. The model excels at generality but does not "excel in... understanding tasks" as claimed.
> >
> > **Q1**: The perception results (especially Tables 2, 4) are lower than some SOTA unified models and specialist models significantly. Is this performance gap an unavoidable cost of the "jack of all trades" unification, or do the authors see a clear path for Jodi to actually surpass some specialist models in understanding tasks?
>
> Thanks for your comments. We have revised the claim of "excel in ... understanding tasks" in the Abstract as suggested, highlighted in red.
>
> We would like to emphasize that the primary goal of our work is to introduce the idea of "joint modeling" for unifying visual generation and understanding tasks, rather than to outperform specialist models on every individual task. Since it is not our focus, we did not make any task-specific designs nor exhaust all task-specific data sources. Therefore, it is reasonable that our Jodi currently lags behind some carefully designed SOTA specialist models on depth and albedo estimation.
>
> Despite this gap, Jodi matches the performance of specialist models and surpasses other unified models on normal estimation and edge detection (Table 4 & 6). In addition, for all controllable generation tasks, Jodi already performs better than or on par with specialist models (Table 2). Therefore, generally speaking, Jodi has already exhibited promising performance across various generation and understanding tasks.
>
> In fact, it typically requires multiple generations of development for a unified model to outperform specialist models. Closely related evidence comes from the development of LLMs, which unify diverse generation and understanding tasks in NLP. Looking back at the early stages of LLMs, such as the representative GPT-3 [1], it was often beaten by specialist models on many tasks, such as ARC Challenge and Winogrande. Nonetheless, subsequent developments, such as GPT-4 [2], demonstrated that LLMs can significantly surpass specialist models. Today, modern LLMs such as GPT-5 [3] are widely used in our daily lives, bringing unprecedented convenience and efficiency to a wide variety of tasks. This evolution trajectory of LLMs indicates that even if unified models initially lag behind specialist models, they have great potential to outperform specialist models and become widely adopted in practical applications.
>
> Returning to our work, we view Jodi as an early implementation of the "joint modeling" idea, aimed at unifying visual generation and understanding. We have verified the feasibility of joint modeling and demonstrated its effectiveness on various controllable generation and understanding tasks. Therefore, we believe that Jodi has strong potential for continuous improvement and development, eventually reaching or even surpassing the performance of specialist models on understanding tasks.
>
> &nbsp;
>
> > **Q2**: What was the sampling strategy for the [G], [C], and [X] roles? Were they sampled uniformly at random for all domains? Or was a curriculum used (e.g., more [C] roles early in training) to stabilize the learning of this complex joint distribution?
>
> Our role sampling strategy is task-centric. We first uniformly sample one task from "joint generation", "controllable generation", and "image perception" with equal probability. Then, we assign roles based on the type of task. For joint generation task, all domains are assigned the role [G], so that the model learns to generate all of them simultaneously. For controllable generation task, where the model is expected to generate images based on any combination of labels, we assign the image domain the role [G], and uniformly assign [C] or [X] to other label domains. For image perception task, the image domain is always assigned [C], while the label domains are uniformly assigned [G] or [X].
>
> &nbsp;
>
> [1] Brown, Tom, et al. "Language models are few-shot learners." Advances in neural information processing systems 33 (2020): 1877-1901.
>
> [2] Achiam, Josh, et al. "Gpt-4 technical report." arXiv preprint arXiv:2303.08774 (2023).
>
> [3] OpenAI. "Introducing GPT-5." https://openai.com/index/introducing-gpt-5/

---

### Official Review · Reviewer_yWSM · 2025-11-02

**Soundness:** 4
**Presentation:** 3
**Contribution:** 4
**Rating:** 6
**Confidence:** 4

**Summary:**

This paper introduces Jodi, a diffusion-based model that tries to unify image generation and image understanding by jointly modeling images and multiple label modalities. With the proposed “Role-Switch” mechanism, the same model can generate images with labels, generate images conditioned on any combination of labels, or predict labels from an input image. It uses a linear diffusion transformer and domain-invariant positional embeddings to keep computation manageable and maintain consistency across domains. The authors also provide a new dataset covering 8 visual domains. Experiments show good performance on both generation and perception tasks and strong scalability. Overall, the idea is intuitive and nicely executed, and the results suggest that a unified approach like this is becoming useful.

**Strengths:**

* The role switch mechanism:  by randomly switching each domain between being generated, used as conditioning, or ignored, the model learns all the key distributions for both image generation and perception at once -- giving one network the flexibility to do many things.

* Paper is well motivated, and generally well written and easy to follow.

* The proposed method uses shared positional embeddings across domains (+ small role tags) so that the model knows which pixels align spatially, making it easier to keep different visual modalities consistent with each other.

* The proposed Joint generation could be very valuable in many downstream applications, such as artistic manipulation or asset usage in the traditional frameworks.

* Experimental results validate the proposed method convincingly.

**Weaknesses:**

* “Extensibility to more domains” is claimed, but it’s unclear how much effort is needed for an entirely new modality.

* Baselines for multimodal generation/understanding might not share the same supervision or label availability.
This needs to be clarified in the paper.

* Randomly switching roles during training might make optimization harder or convergence slower, especially as domains increase.

* Even with linear attention, jointly modeling 8+ domains could still be memory-intensive and slow for large resolutions.

* The “multiple label domains” are pseudo labels (depth, normals, etc.), so errors/noise in these domains may propagate through the unified model.

**Questions:**

* How well does the method scale to more detailed label domains (e.g., full-class/instance segmentation or dense keypoints) without redesign?

* What are the failure cases on real or diverse datasets, especially with human subjects? Although the limitations are discussed, the failure cases are not provided (neither in the main paper nor in the supplementary).

* Can the authors provide per-domain trade-offs -- do any tasks degrade notably compared to specialist methods?
 The results currently reported appear not to be on fair ground comparisons.

---

> ### Author Response · Authors · 2025-11-24
> **Response to reviewer yWSM (Part 1/2)**
>
> We sincerely thank Reviewer yWSM for carefully reading our paper and giving valuable feedback.
>
> &nbsp;
>
> > **W1**: “Extensibility to more domains” is claimed, but it’s unclear how much effort is needed for an entirely new modality.
>
> We have added additional remarks on the domain extension experiments in Appendix Section E of the revised paper, highlighted in red. We fine-tuned our Jodi model on only one RTX A6000 GPU with a reduced batch size of 4 and a learning rate of $1\times 10^{-5}$. To illustrate the efficiency, we present the joint generation results throughout the fine-tuning process in Figure 15, comparing with directly training from Sana. It is clear that the newly added domains converge within 2,000 fine-tuning steps (around 10 GPU hours), whereas training from Sana still yields unsatisfactory results even after 5,000 steps. This demonstrates the effectiveness and efficiency of our Jodi in extending new domains.
>
> &nbsp;
>
> > **W2**: Baselines for multimodal generation/understanding might not share the same supervision or label availability. This needs to be clarified in the paper.
>
> Thank you for your suggestion. We have added the detailed training setting and clarification in Section 4.1 and Table 1 of the revised paper, highlighted in red. Table 1 is attached below for your reference.
>
> | Method | Base Model | \# Parameters | Dataset |
> |---|---|---|---|
> | OmniGen | Phi-3 | 3.8B | X2I (100M) |
> | PixWizard | Lumina-Next-T2I | 2B | PixWizard (30M) |
> | OneDiffusion | (from scratch) | 2.8B | One-Gen (75M) |
> | Jodi (ours) | Sana | 1.6B | Joint-1.6M (200K) + GT labels (90K) |
>
> &nbsp;
>
> > **W4**: Even with linear attention, jointly modeling 8+ domains could still be memory-intensive and slow for large resolutions.
>
> As shown in Figure 12 in the Appendix, increasing the number of domains from 2 to 8 results in only a modest memory increase from 37GiB to 41GiB. Regarding time consumption, we prioritize effectiveness over efficiency since this work aims to validate the feasibility of our "joint modeling" idea for unifying visual generation and understanding tasks. We will explore optimizing training speed and inference latency in future work.
>
> &nbsp;
>
> > **W5**: The “multiple label domains” are pseudo labels (depth, normals, etc.), so errors/noise in these domains may propagate through the unified model.
>
> We fully agree with your concerns, which we have also acknowledged in Section 3.3 of our paper. To solve this problem, we incorporate datasets with ground-truth labels during training. We have an ablation study on the effect of ground-truth labels in Table 14 of the Appendix, showing that ground-truth labels significantly improve performance on perception tasks, validating their necessity.

---

> ### Author Response · Authors · 2025-11-24
> **Response to reviewer yWSM (Part 2/2)**
>
> > **Q1**: How well does the method scale to more detailed label domains (e.g., full-class/instance segmentation or dense keypoints) without redesign?
>
> We have added experiments on full-class semantic segmentation in Appendix Section D of the revised paper. To encode the classes into RGB colors, we use both the official color mapping and a remapping strategy that ensures the colors are well distributed across the RGB space. Please refer to Figure 13 for an intuitive illustration. To investigate the performance of our method on the semantic segmentation task with 150 classes, we replace the 12-class segmentation domain in the pretrained Jodi model with the 150-class segmentation domain and fine-tune the model on the ADE20K dataset for 20K steps. The quantitative and qualitative results are shown in Table 9 (attached below for your reference) and Figure 14. Jodi outperforms the previous unified model, PixWizard, even when using the official color mapping. Furthermore, our color remapping strategy significantly improves performance, highlighting the importance of maintaining sufficient separability between classes. However, our results still lag behind those of specialist models. We attribute this gap to the limitations of the RGB space, where distances between colors do not correspond to semantic similarity. For example, semantically related classes "car", "van", and "truck" are mapped to highly distinct colors, while unrelated classes such as "plant" and "sidewalk" are mapped to visually similar colors. This distorted color-semantic relationship introduces unnecessary learning difficulty. In future work, we plan to explore a more suitable representation space beyond RGB for the segmentation domain.
>
> | Method | mIoU |
> |---|---|
> | UniFormer | 44.4 |
> | OneFormer | 57.3 |
> | PixWizard | 7.0 |
> | Jodi (official color mapping) | 11.9 |
> | Jodi (our color remapping) | 17.3 |
>
> &nbsp;
>
> > **Q2**: What are the failure cases on real or diverse datasets, especially with human subjects? Although the limitations are discussed, the failure cases are not provided (neither in the main paper nor in the supplementary).
>
> We include failure cases in Figure 16 in the revised paper. Since we mainly focus on the unification of generation and understanding tasks, image quality is not prioritized. We did not specifically curate data for complex scenes, such as images with multiple objects or human subjects. Also, we did not apply high-quality fine-tuning or preference optimization techniques that are commonly used in T2I models. We believe that this issue could be alleviated by collecting more high quality data and applying advanced techniques, which we leave for future improvement.
>
> &nbsp;
>
> > **Q3**: Can the authors provide per-domain trade-offs -- do any tasks degrade notably compared to specialist methods? The results currently reported appear not to be on fair ground comparisons.
>
> As shown in Table 3 & 5, our Jodi currently lags behind specialist methods on depth and albedo estimation. This is because specialist models often rely on task-specific designs and carefully curated datasets. For example, Lotus [1] proposes to predict the depth maps instead of noise or velocity fields, and reformulates the diffusion process into a single-step procedure. It also incorporates Virtual KITTI [2] as an important source of outdoor scene data. In contrast, since our work focuses on unifying a wide range of tasks, we use a general design and do not exhaust all possible data sources.
>
> However, for other perception tasks including normal estimation and edge detection, Jodi reaches the performance of specialist models (Table 4 & 6). Moreover, for controllable generation tasks, Jodi performs better than or on par with specialist models (Table 2). Generally speaking, our Jodi shows strong performance across a variety of generation and understanding tasks.
>
> Finally, we would like to emphasize that our aim is not to outperform specialist models on every task, but to introduce the idea of "joint modeling" for unifying visual generation and understanding tasks. Given the results in our paper, we believe that Jodi is a promising starting point that can be further improved with additional data, computational resources, and engineering efforts.
>
> &nbsp;
>
> [1] He, Jing, et al. "Lotus: Diffusion-based Visual Foundation Model for High-quality Dense Prediction." The Thirteenth International Conference on Learning Representations.
>
> [2] Cabon, Yohann, Naila Murray, and Martin Humenberger. "Virtual kitti 2." arXiv preprint arXiv:2001.10773 (2020).

---

### Author Response · Authors · 2025-12-04
**Rebuttal Summary (Part 1/2)**

We sincerely thank the Area Chair and all reviewers for their time, efforts, and valuable feedback throughout the review and rebuttal process. To facilitate the evaluation of our paper, we would like to provide an overview of our contributions, a summary of the reviews, a list of key resolutions, and revisions made during the rebuttal period.

&nbsp;

### **Our Contributions**

- First, we reveal a theoretical perspective showing that the *joint distribution $p(x,y)$ inherently encodes the interdependence between generation tasks (formulated as $p(x|y)$) and understanding tasks (formulated as $p(y|x)$).*

- Guided by this perspective, we propose **jointly modeling** the image domain and the label domains to achieve the unification of visual generation and understanding. To realize this idea, we design a novel Role-Switch mechanism that allows a single diffusion model to cover diverse distributions, including three most typical ones: 1) $p(x,y^1,y^2,\cdots)$, joint generation, where the model simultaneously generates both the image and the corresponding labels of different domains; 2) $p(x|y^1,y^2,\cdots)$, controllable generation, where the images are generated conditioned on any combination of the label domains; 3) $p(y^1,y^2,\cdots|x)$, image perception, where the model accepts an input image and predicts multiple labels at once.

- Moreover, to reduce computational complexity and encourage spatial alignment across visual domains, we adopt masked linear attention and propose domain-invariant positional embeddings.

- Additionally, we introduce the Joint-1.6M dataset to facilitate further research in this area. It contains 200K high-quality images collected from public sources, automatic labels for 7 visual domains, and LLM-generated captions.

&nbsp;

### **Strength Summary**

We sincerely appreciate the reviewers for recognizing the strengths and contributions of our paper, including:

- **Core Idea and Motivation**: Our probabilistic perspective of connecting generation tasks $p(x|y)$ and understanding tasks $p(y|x)$ via the joint distribution $p(x,y)$ is described as a "Principled and Elegant Framework ... statistically elegant" (**LQZ6**) and "I like the probabilistic perspective on joint modeling ... provides an intuitive motivation" (**Qnvb**).

- **Implementation and Architecture**: Our proposed Role-Switch mechanism is regarded as "a clever and direct implementation" (**LQZ6**), achieving "Flexible Control" (**qawp**) and "giving one network the flexibility to do many things" (**yWSM**). The linear attention and domain-invariant positional embeddings are praised as "smart architectural choices ... solid supporting contributions that are well-motivated and essential" (**LQZ6**) and "making it easier to keep different visual modalities consistent with each other" (**yWSM**).

- **Practical Value**: Our model is considered "very valuable in many downstream applications" (**yWSM**) and offers "flexibility for creative applications" (**qawp**). The proposed Joint-1.6M dataset is also highlighted as "a valuable contribution to the community" (**LQZ6**).

- **Experimental Results**: Reviewers note that the "Experimental results validate the proposed method convincingly" (**yWSM**) and that "Visual results clearly show that your method works" (**Qnvb**).

---

> ### Author Response · Authors · 2025-12-04
> **Rebuttal Summary (Part 2/2)**
>
> ### **Key Concerns and Our Responses**
>
> During the rebuttal, we carefully addressed all concerns raised by the reviewers. Key concerns and our responses are summarized below:
>
> **Performance gap compared with specialist models on some perception tasks (yWSM, LQZ6)**
>
> We would like to emphasize that the primary goal of our work is to introduce the idea of "joint modeling" for unifying visual generation and understanding tasks, rather than to outperform specialist models on every individual task. Since it is not our focus, we did not make any task-specific designs nor exhaust all task-specific data sources. Therefore, it is reasonable that our Jodi currently lags behind some carefully designed specialist models on depth and albedo estimation.
> Nonetheless, for surface normal estimation and edge detection, Jodi matches the performance of specialist models and surpasses other unified models. Furthermore, for all controllable generation tasks, Jodi already performs better than or on par with specialist models. Generally speaking, Jodi has already exhibited promising performance across various generation and understanding tasks.
>
> **Importance of unification and the potential of our method (yWSM, LQZ6, Qnvb)**
>
> First, we illustrate the importance of unification by drawing examples from the development of LLMs, which unify diverse generation and understanding tasks in NLP. Looking back at the early stages of LLMs, such as the representative GPT-3, it was often beaten by specialist models on many tasks, such as ARC Challenge and Winogrande. Nonetheless, subsequent developments, such as GPT-4, demonstrated that LLMs can significantly surpass specialist models. Today, modern LLMs such as GPT-5 and Gemini-3 are widely used in our daily lives, bringing unprecedented convenience and efficiency to a wide variety of tasks. This evolution trajectory of LLMs indicates that even if unified models initially lag behind specialist models, they have great potential to outperform specialist models and become widely adopted in practical applications.
>
> Returning to our work, we view our Jodi as an early implementation of the "joint modeling" idea, aimed at unifying visual generation and understanding. Our paper has verified the feasibility of joint modeling and demonstrated its effectiveness on various controllable generation and understanding tasks. Therefore, we believe that Jodi has strong potential for continuous improvement and development, eventually reaching or even surpassing the performance of specialist models on all kinds of tasks.
>
> **Practical value of our method (Qnvb)**
>
> From a practical standpoint, our Jodi brings great convenience in deployment. Users no longer need to manage multiple environments, download separate model weights, and run different inference pipelines for each task. In addition, Jodi can predict multiple labels simultaneously in a single sampling process, while using specialist models requires a sequence of separate inference processes. Furthermore, Jodi can perform joint generation, which provides practical value for data synthesis and other downstream applications, as noted by *Reviewer yWSM: "The proposed Joint generation could be very valuable in many downstream applications, such as artistic manipulation or asset usage in the traditional frameworks."*
>
> &nbsp;
>
> ### **Revision List**
>
> We revised our paper according to the reviewers' comments, with all modifications highlighted in red. These revisions include:
>
> - We revised the claim regarding understanding task performance in the Abstract. (**LQZ6**)
>
> - We added discussions of 3DIS and 3DIS-FLUX in Section 2. (**qawp**)
>
> - We clarified the detailed training setting of the other unified methods in Section 4.1 and Table 1. (**yWSM**)
>
> - We clarified the meaning of bold numbers in the tables. (**qawp**)
>
> - We added experiments for the 150-class segmentation task with two types of color mappings, and discussed limitations of RGB space in detail in Appendix Section D. (**yWSM**)
>
> - We added an experiment in Appendix Section E demonstrating the efficiency of extending our model to new domains. (**yWSM**)
>
> - We added T2I evaluation results and showed failure cases in Figure 16, and provided analysis in the responses. (**yWSM, qawp**)
>
> &nbsp;
>
> We once again thank the Area Chair and all reviewers for carefully reviewing our paper and providing valuable feedback. We hope that our responses and revisions have addressed the reviewers' concerns and further improved the quality of our work.

---

### Meta-Review · Area_Chair_RjcU · 2025-12-15

**Summary:**

The reviewers mainly concern about the 1) performance gap between the unified model and specialist models, 2) the practical value of the framework, and 3) specific evaluation metrics. For example, for (1), Reviewer LQZ6 mentioned that Jodi performs worse than SOTA specialist models on albedo estimation and depth estimation. For (2), Reviewer Qnvb questioned why users would adopt Jodi instead of using pre-trained specialized models to achieve SOTA performance. For (3), Reviewer qawp pointed out that there is no T2I evaluation.

**Reviewer Concerns:**

The authors addressed many of the concerns, including the practical value, extension to new domains, and T2I evaluation. However, there are unresolved concerns, including the performance gap, generation quality, and segmentation quality.

**Reviewer Scores:**

This paper receives initial ratings of (2, 4, 6, 6). Reviewer Qnvb, who corresponds to the 2, mentioned that the score has been raised. Other reviewers have no response. Based on the rebuttal, the AC anticipates a score of (4, 4, 6, 6), given the aforementioned outstanding concerns.

---

### Decision · Program_Chairs · 2026-01-26

Reject